# Global diversity dynamics in the fossil record are regionally heterogeneous

Joseph T. Flannery-Sutherland [1✉], Daniele Silvestro [2,3,4] & Michael J. Benton [1]

Global diversity patterns in the fossil record comprise a mosaic of regional trends, under-pinned by spatially non-random drivers and distorted by variation in sampling intensity through time and across space. Sampling-corrected diversity estimates from spatially-standardised fossil datasets retain their regional biogeographic nuances and avoid these biases, yet diversity-through-time arises from the interplay of origination and extinction, the processes that shape macroevolutionary history. Here we present a subsampling algorithm to eliminate spatial sampling bias, coupled with advanced probabilistic methods for estimating origination and extinction rates and a Bayesian method for estimating sampling-corrected diversity. We then re-examine the Late Permian to Early Jurassic marine fossil record, an interval spanning several global biotic upheavals that shaped the origins of the modern marine biosphere. We find that origination and extinction rates are regionally heterogenous even during events that manifested globally, highlighting the need for spatially explicit views of macroevolutionary processes through geological time.

[1] School of Earth Sciences, University of Bristol, Bristol, UK. [2] Department of Biology, University of Fribourg, Fribourg, Switzerland. [3] Department of Biological and Environmental Sciences, University of Gothenburg, Gothenburg, Sweden. [4] Global Gothenburg Biodiversity Centre, Gothenburg, Sweden. ✉email: jf15558@bristol.ac.uk

The fossil record is our only empirical sample of past biodiversity, providing a critical resource for understanding macroevolutionary and macroecological processes in deep time[1]. Numerous abiotic and biological drivers have been proposed to explain apparent patterns of fossil diversity[2], but it has long been recognised[3] that these patterns are heavily distorted by uneven sampling intensity through time from geological biases that affect the temporal distribution of fossils and formations[4–6], differing preservation potential across organisms and environments[7], and heterogeneity in collection practice, reporting and even geopolitics[8,9]. These factors are often interlinked and are also geographically variable in their manifestation[10,11]. Therefore, the known fossil record is not only an incomplete sample of the total fossil record (itself a biased fraction of past diversity as a whole), but that incompleteness is also inconsistent through time and across space[12].

Significant attention has been devoted to correcting diversity estimates for temporal variation in sampling intensity[13,14], but it has also been demonstrated that variation in the palaeogeographic distribution of the fossil record through time imposes an equally severe distortion on patterns of diversity even after correction for uneven sampling intensity[2,15–17]. Furthermore, fossil diversity is itself geographically variable due to the spatially nonrandom distribution of factors influencing species richness, for example the locations of reefs and epeiric seaways, or climatically structured latitudinal diversity gradients[17,18]. Recent studies of global fossil diversity have calculated pointwise diversity estimates from temporally standardised, spatially-even subsamples of fossil data[2,16,17], allowing the mosaic of global diversity to be decomposed into its regional components while accounting for the distortion induced by spatial sampling bias[12]. Focusing on diversity alone, however, is limiting as it is ultimately a dynamic product of origination and extinction rates[19]. Standing diversity, as determined by these rates at any point in time, then interacts with a spatiotemporally variable sampling rate to produce the fossil record. A drop in apparent diversity may result from a drop in origination or sampling rate just as much as from an increase in extinction rate, while a relatively flat diversity trajectory could mask cryptic phases of turnover resulting from concurrent pulses of origination and extinction. A few studies have used geographically restricted datasets to gain regional views of diversification rates through time[20,21], but there are currently no methods to generate fossil datasets that are spatially uniform through time, and this seriously hinders investigation of diversity dynamics at different spatial scales and between different geographic regions.

In this paper we present a subsampling algorithm to produce spatially-standardised fossil occurrence datasets which remain geographically consistent through time, along with a method of calculating sampling-corrected diversity in a Bayesian framework to complement the inference of sampling-corrected origination, extinction and preservation rates in the software packages PyRate and LiteRate[22–25]. We apply these methods to a composite dataset of marine fossil occurrences spanning the Late Permian to Early Jurassic, an interval characterised by a dramatic backdrop of interlinked palaeogeographic shifts[26], climatic fluctuations[26] and three extinction events: the Permo-Triassic mass extinction (PTME)[27]; the Carnian Pluvial Episode (CPE)[28]; and the Triassic-Jurassic mass extinction (TJME)[29], alongside a series of other less well understood biotic upheavals (e.g. the Smithian-Spathian Event[30] and the Mid-Norian Climate Shift[31]). We find that global trends are heavily biased by the regional distribution of the fossil record and regional diversity dynamics themselves are strongly heterogenous even during supposedly global biotic events, indicating that global trends are not simply an upscaling of regional processes. This regional variability reflects the unique biogeographic histories of each study region, demonstrating the importance of geographic context in the assembly and transformation of biodiversity through deep time and highlighting how our view of the history of global diversity remains biased by the uneven spatial distribution of the fossil record.

## Results

**Spatial standardisation**. We captured regional samples of fossil occurrence data (West Circumtethys, East Circumtethys, North Panthalassic, South Panthalassic, Boreal, Tangaroan) using sliding spatial windows, binned the data spatially using a hexagonal grid (Fig. 1) and standardised the spatially binned extent of the data through time by its longitude–latitude range and minimum spanning tree (MST) length. The resulting samples of fossil occurrence data geographically consistent through time and free from spatial sampling biases which can substantially distort trends in apparent diversity. When coupled with diversity estimation methods which correct for heterogenous sampling between time bins, our workflow permits estimation of diversity dynamics unaffected by spatiotemporal sampling bias, allowing regional diversity dynamics to be interrogated.

Our spatial standardisation workflow successfully reduced variance in MST length and longitude–latitude range whilst enforcing a consistent geographic distribution of data through time in each region, although the degree of reduction is dependent on the dataset and target extent, with a noticeable increase in standardisation efficacy with increasing region size (Table 1). The standard deviations in realised MST length relative to target MST length for the large Circumtethyan regions are all less than 1.4% after spatial standardisation as a target length suitable for all bins could be chosen. By comparison, the standard deviation around the target rises for the smaller North Panthalassic region as we chose a target that improved data retention for the vast majority of the bins (Early Triassic–Early Jurassic), but which was significantly higher than the unstandardised spatial extent of the data in the Late Permian, resulting in greater variance when the full time span is considered (Table 1; Fig. S5). Similarly, the South Panthalassic region is well standardised from the Norian to the Early Jurassic, allowing the signal of the TJME to be scrutinised, despite the vastly reduced spatial extent at the start of the dataset (Table 1; Fig. S7). The constraints imposed by standardisation for both spatial metrics are also apparent in the Tangaroan region, where MST standardisation is reasonably effective throughout the Triassic but declines in quality when longitude–latitude standardisation is first applied (Table 1; Fig. S6).

Prior to standardisation, the relationship between region-level (RL) diversity by coverage-based rarefaction and spatial extent is significant for multiple regions across the measured quorum levels (Fig. 2). After standardisation, significant correlations with RL diversity are broadly eliminated but some are present for comparisons which were previously insignificant. Regardless, we were able to produce at least one dataset for each region where RL diversity at each quorum level showed no significant correlations with spatial extent, with the exception of the North Panthalassic region. Here, significant Spearman correlations were still present at some quorum levels (Table S54), but the lack of a consistent correlation across quorum levels, along with their weak statistical significance, suggests that the apparent relationships are not robust. Further, all correlations were rendered insignificant when the two Late Permian bins were excluded from the analyses (Fig. 2), indicating that the remainder of the dataset is otherwise well standardised (Table 1).

**Probabilistic origination, extinction and diversity**. Diversification, speciation and extinction rates and probabilistic diversity

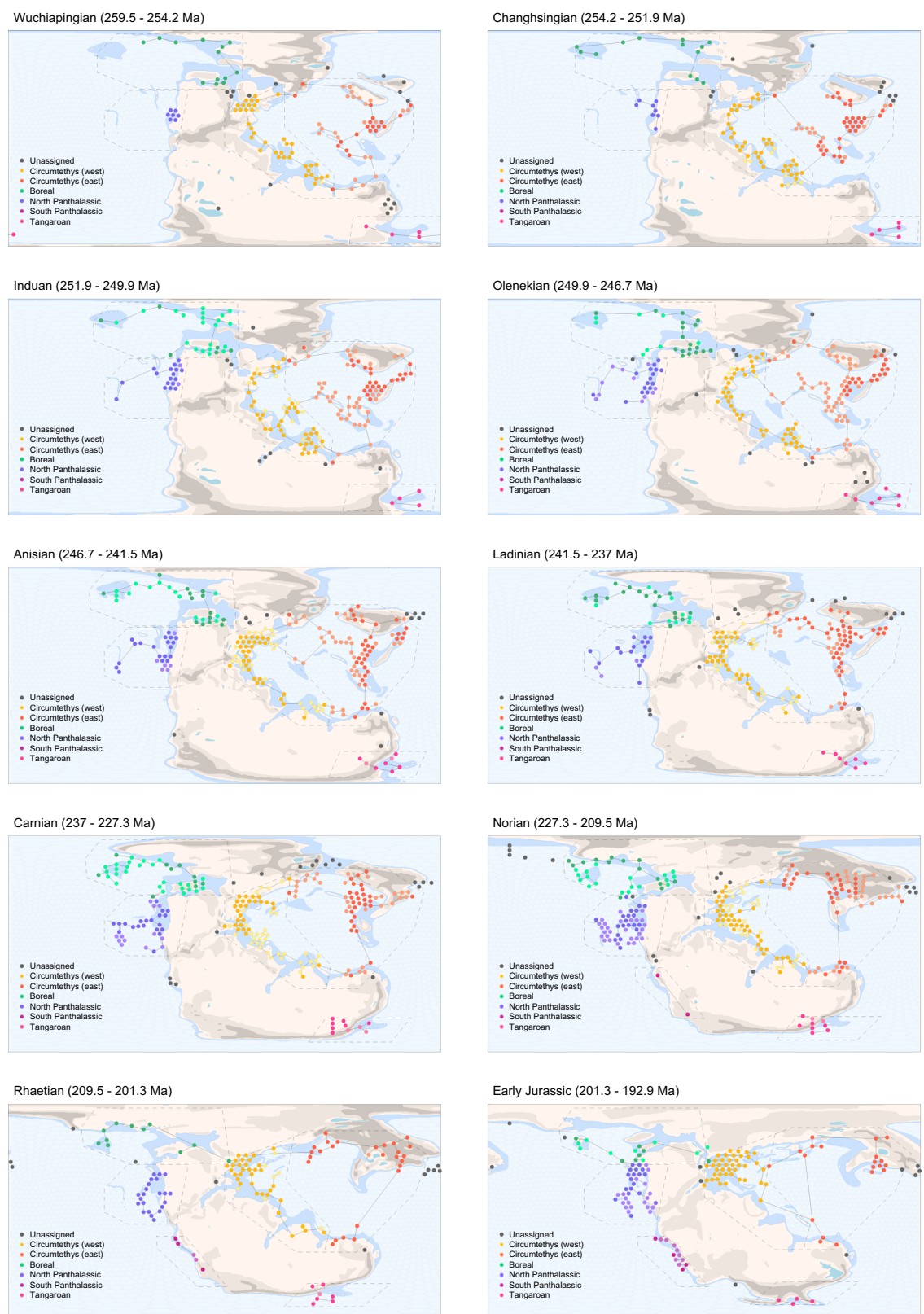

**Fig. 1 Spatial standardisation for the Late Permian to Early Jurassic (259.5–192.9 Ma).** Geographically consistent temporal transects of fossil data were captured using sliding spatial windows which demarcate the bounds of biogeographically important sampling regions through our study interval, then spatially subsampled to give a consistent minimum spanning tree length (MST) in each time bin. Spatially-standardised datasets from each region were used to estimate diversity dynamics corrected for heterogenous sampling through time and across space. Dotted black lines demarcate the sliding spatial windows defining each sampling region, solid black lines their MSTs, coloured points the occupied grid cells comprising the MST, and points with the darker hue those retained after MST standardisation.

**Table 1 Target extent and efficacy for minimum spanning tree (MST) and longitude–latitude standardisation.**

| | Target extent | Unstandardised SD (%) | Standardised SD (%) |
|---|---|---|---|
| MST length (km) | | | |
| West Circumtethys | 23,000 | 16.7 | 1.2 |
| East Circumtethys | 23,000 | 33.7 | 1.3 |
| Boreal | 10,000 | 35.6 | 9 |
| North Panthalassic[a] | 12,000 | 29.6 | 3.4 |
| North Panthalassic | 12,000 | 49.9 | 28.6 |
| Tangaroan | 3000 | 29.5 | 22.6 |
| South Panthalassic[b] | 2500 | 42.7 | 12.2 |
| South Panthalassic | 2500 | 88.7 | 70.1 |
| Longitude range (°) | | | |
| West Circumtethys | 60 | 15.8 | 5.3 |
| East Circumtethys | 95 | 5 | 3.6 |
| Boreal | 30 | 15.2 | 9.1 |
| North Panthalassic[a] | 30 | 34.9 | 34.8 |
| North Panthalassic | 30 | 53.2 | 25.6 |
| Tangaroan | 30 | 35.1 | 14.6 |
| South Panthalassic[b] | 90 | 9.4 | 9.4 |
| South Panthalassic | 90 | 42.4 | 42.1 |
| Latitude range (°) | | | |
| West Circumtethys | 75 | 16.4 | 14.5 |
| East Circumtethys | 90 | 12.7 | 11.6 |
| Boreal | 35 | 14.4 | 16.7 |
| North Panthalassic[a] | 30 | 16.3 | 15.4 |
| North Panthalassic | 30 | 31 | 23.4 |
| Tangaroan | 10 | 28.7 | 20.1 |
| South Panthalassic[b] | 35 | 15.2 | 15.2 |
| South Panthalassic | 35 | 46.8 | 45.6 |

*SD* standard deviation relative to target.
[a]Early Triassic to Early Jurassic only.
[b]Rhaetian to Early Jurassic only.

trajectories clearly differ between sampling regions even during well documented global events (Figs. 3–5). The signal of the PTME is clear cut in the Circumtethyan and North Panthalassic regions, but the onset of elevated extinction rates occurs a couple of million years earlier in the latter (Fig. 5F) and may reflect the age uncertainty of the fossil occurrences. An increase in extinction rate is also present in the Tangaroan region, although Bayes Factor support for an increase in extinction rate here is barely positive rather than strong (Fig. S31). In the Boreal region, however, there is high uncertainty in the magnitude of the extinction rate increase at the Permo-Triassic boundary and the median origination rate remains consistently higher than the median extinction rate (Fig. 5B), producing a subdued extinction signal with very little change in diversity. Further paroxysms in extinction rates are present in the North Panthalassic, Boreal and Tangaroan regions throughout the Early Triassic, again with positive rather than strong support in the latter. There is a clear spike in extinction rate in West Circumtethys at the CPE (Fig. 5C), along with more subdued increases in the Boreal and East Circumtethys regions (Fig. 5B, D). Elsewhere, extinction rates show little change through the CPE, while elevated extinction rates instead occur at the end of the Carnian in the Circumtethyan, Boreal and North Panthalassic regions (Fig. 5B–D, F). Distinct extinction signals are present in all regions at the end of the Triassic, but this is somewhat reduced in West Circumtethys due to a concurrent spike in origination rate (Fig. 5C), while in the Tangaroan region the rate shift significance is again merely positive rather than strongly supported.

Shifts in origination and extinction rate occur frequently throughout the duration of each region, with strong support for their statistical significance. Away from major extinction events where there are clear shifts in the median rate, the majority of these shifts represent minor fluctuations in the background extinction rate or periods where sharp rate changes are inferred but with high uncertainty on their magnitude and timing.

Probabilistic diversity also displays marked short-term fluctuations (Fig. 4), punctuated by sharp peaks and crashes marking major periods of biotic turnover where concurrent disparities between extinction and origination rates (i.e. a sharp change in net diversification rate) may be noted (Figs. 3 and 5).

**Turnover.** As with region-level diversity, within-region turnover shows regional differences in both the magnitude and pattern of dissimilarity through time (Fig. 4). The most pronounced shifts in turnover occur in the Early Triassic in the aftermath of the PTME in all regions, but even here there are distinct regional differences. Turnover spikes occur across the Permo-Triassic boundary in all regions, aside for the Tangaroan where the spike is in the Olenekian rather than in the Induan. West and East Circumtethys show comparable trends through the Late Permian to Carnian, and in both cases turnover throughout the Middle to Late Triassic is lower compared to the Early Triassic. From the Norian onwards, however, West Circumtethys shows steadily greater dissimilarity through time into the Sinemurian, while dissimilarity steadily declines in East Circumtethys before spiking across the Triassic-Jurassic boundary. In the Boreal and North Panthalassic regions, more prominent changes in turnover occur throughout the Late Triassic, and both regions show generally greater turnover than in the Circumtethyan regions. Dissimilarity through time is also more pronounced in the Tangaroan and South Panthalassic regions, which may reflect the impact of reduced sampling in both regions leading to greater incompleteness.

## Discussion

While our workflow is successful in minimising spatial bias, its utility is potentially restricted to large geographic samples and may not scale to smaller regions or clades; this is because

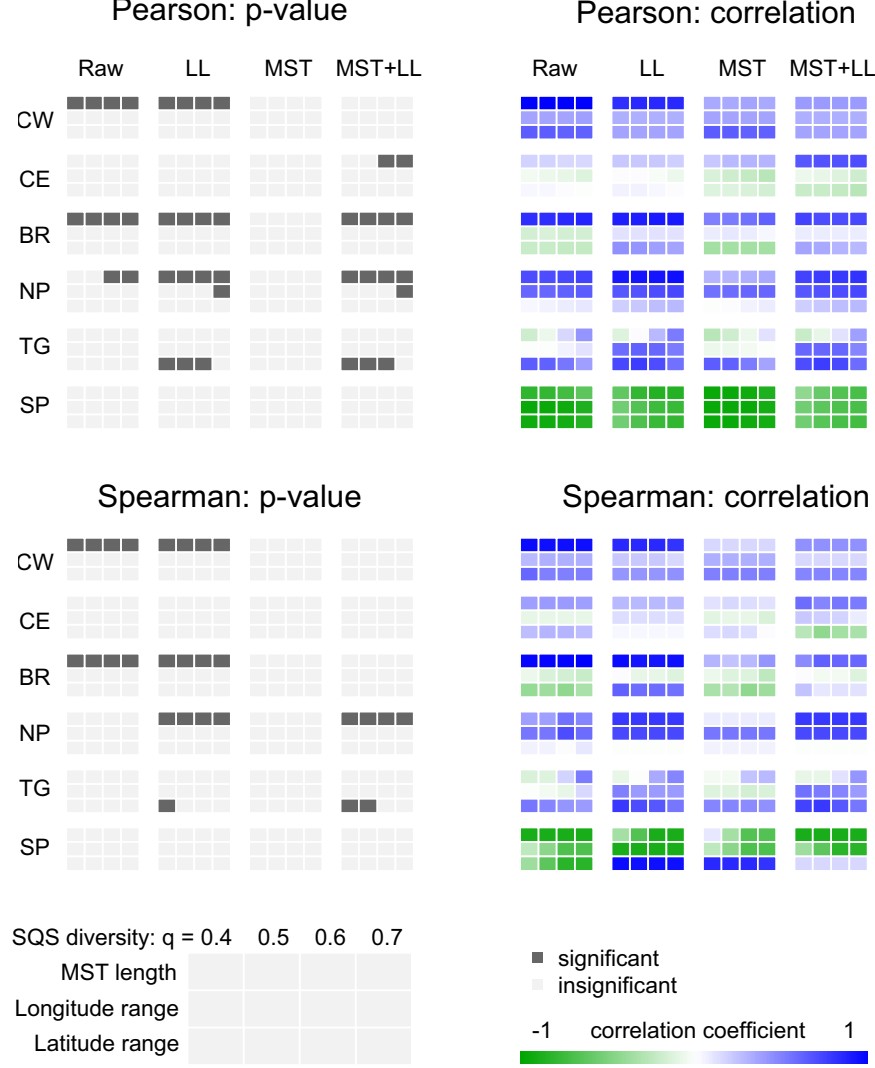

**Fig. 2 One-tailed Pearson and Spearman correlation tests between sampling-corrected diversity and spatial extent under each spatial standardisation treatment, for each sampling region, with false discovery correction.** Without any spatial standardisation (raw), spatial sampling extent frequently shows significant relationships with diversity from shareholder quorum subsampling (SQS) across several quorum (q) levels. Standardisation by minimum spanning tree length (MST) is the most effective means of mitigating this bias, compared to standardisation of longitude–latitude extent (LL). CW West Circumtethys, CE East Circumtethys, BR Boreal, NP North Panthalassic, TG Tangaroan, SP South Panthalassic. For the North Panthalassic region, the Permian was omitted from the time series correlations as an unavoidable increase in spatial sampling extent which otherwise resulted in significant Spearman correlations in longitude and latitude extent at some quorum levels. Exact p-values and correlation coefficients are available in Supplementary Tables 1–64.

increasing spatial or taxonomic granularity would increase the patchiness of sampling through time. Instead, local stratigraphic sections will continue to provide the data required to analyse diversity dynamics at local scales with high temporal resolution. As with the choice of sliding window geometry, the choice of target standardisation extent is dependent on multiple factors, including the availability of data in the initial subsample and potential trade-offs between the length contributed to the MST by its component grid cells versus the amount of data contained by those grid cells. Consequently, there may be scope to develop a Pareto-optimal solution to our subsampling workflow using multi-criterion MSTs (i.e. MSTs that are constructed to satisfy multiple dataset properties in a trade-off) to optimise spatial extents and maximise data retention simultaneously, although this is beyond the bounds of this paper. Demarcating spatial regions using sliding windows is subjective, but it might be possible to identify spatial regions more objectively using

network approaches that detect biogeographic continuity through time[32–34].

Prior to standardisation, significant correlations between spatial extent and diversity corroborate previous findings that variation in the former distorts the latter[2,17]. Not all correlations were significant, however, suggesting that at a regional scale the otherwise strong relationship noted at the global level[2] begins to decouple. Nonetheless, spatial variation in a fossil dataset will still affect measured diversity, even if the net changes in diversity and spatial extent are uncorrelated and so it remains important to reduce this spatial variation to isolate true origination and extinction rates. Significant correlations between diversity and spatial extent after standardisation are unexpected, but these cases are infrequent and may be spurious given that spatial variation is heavily curtailed, substantially reducing its impact on diversity-through time.

The limited degree of qualitative change between rate curves compared to diversity curves for each data standardisation

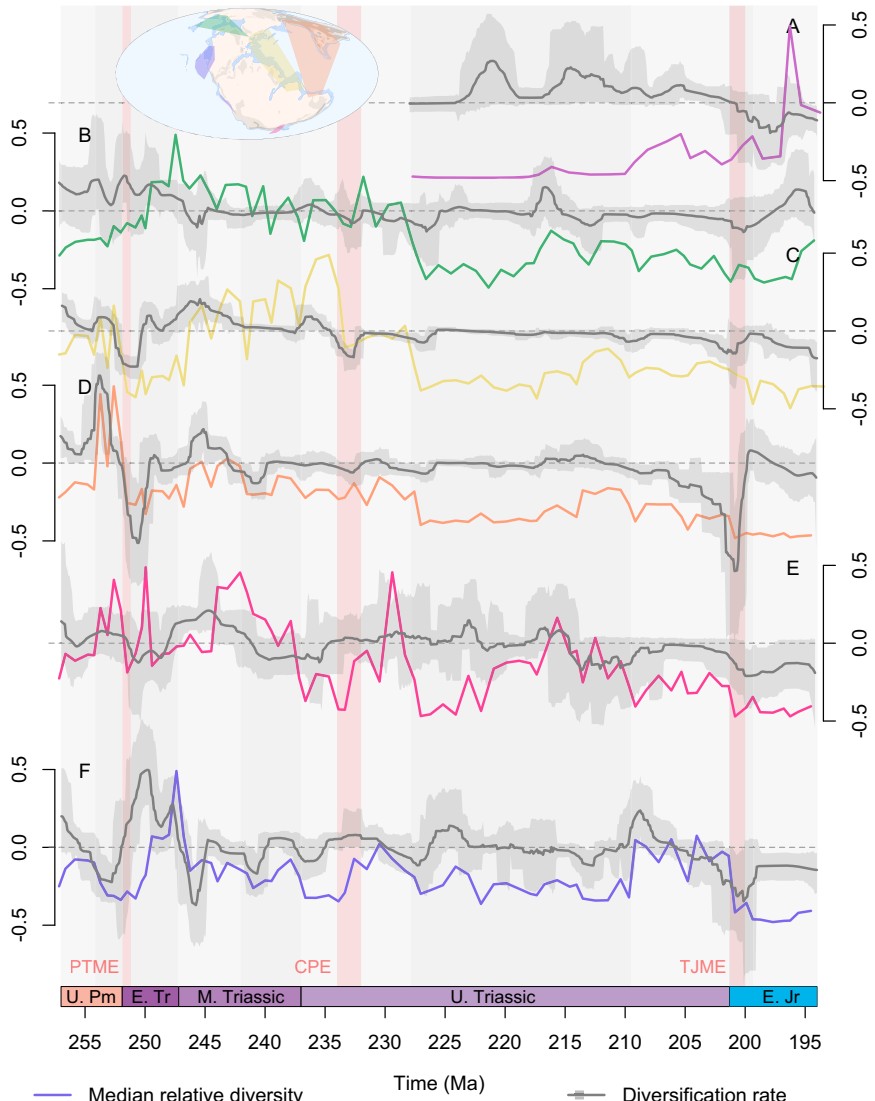

**Fig. 3 Regional probabilistic median diversification rates, their 95% highest posterior densities, and median probabilistic diversity curves through the Late Permian to Early Jurassic (259.5–192.9 Ma).** Diversification rates and the diversity curves resulting from these rates differ markedly between sampling regions, demonstrating that the accumulation of biodiversity is heterogenous around the globe through our study interval, even during globally pervasive biological upheavals like the end-Permian mass extinction (PTME), Carnian Pluvial Episode (CPE), or end-Triassic mass extinction (TJME). Displayed results are derived from PyRate and mcmcDivE analyses applied to datasets standardised by minimum spanning tree length, meaning that the results are unbiased by heterogenous sampling through time and across space. Y-axes are not provided for diversity as these values are rescaled so that the highest point in each time series is equal to 1 (see Fig. 2 for axes and confidence intervals). **A** South Panthalassic. **B** Boreal. **C** West Circumtethys. **D** East Circumtethys. **E** Tangaroan. **F** North Panthalassic. Source data are provided as a Source Data file.

treatment (e.g. Boreal, Fig. S40) shows that taxonomic ranges in the fossil record are more robust to spatial sampling bias than standing diversity. The quantitative differences between standardisation treatments, however, demonstrate that spatial sampling bias still affects origination and extinction rates. As spatial sampling extent grows so too does the likelihood that some of the increasing number of fossil occurrences sampled will be of new taxa, inflating observed fossil diversity under the species-area effect[2]. We propose that greater spatial extent additionally increases the likelihood that some of the sampled occurrences will represent the FADs and LADs for their lineages. Thus, as spatial extent fluctuates through time, fluctuation in the capture of FADs and LADs will distort origination and extinction rates even if the FADs and LADs themselves are relatively unbiased. Preservation rate, however, may still vary independently of the spatial extent of a sample (for example, the influence of geographically localised,

but taxonomically well-sampled lagerstätten) and so will continue to distort FADs and LADs, along with their potential for discovery as the spatial extent of the fossil record fluctuates. Further, while a taxon has an absolute FAD and LAD, these values may vary regionally through heterogeneity in the time required for a taxon to disperse from its point of origin into a new region and in the timing of its extirpation. Thus, the fossil record may fail to capture absolute taxon FADs and LADs through fluctuation in spatial sampling extent and instead only preserve regional originations and extirpations, biasing individual taxon ranges.

Not only does spatial sampling bias affect rate estimates, but spatial variation in sampling intensity also biases the composition of the 'global' fossil record. The differences between diversity from the total Circumtethyan dataset to those from its eastern and western subdomains demonstrates dissonance in diversity dynamics at different spatial scales (Fig. 6). Data from West

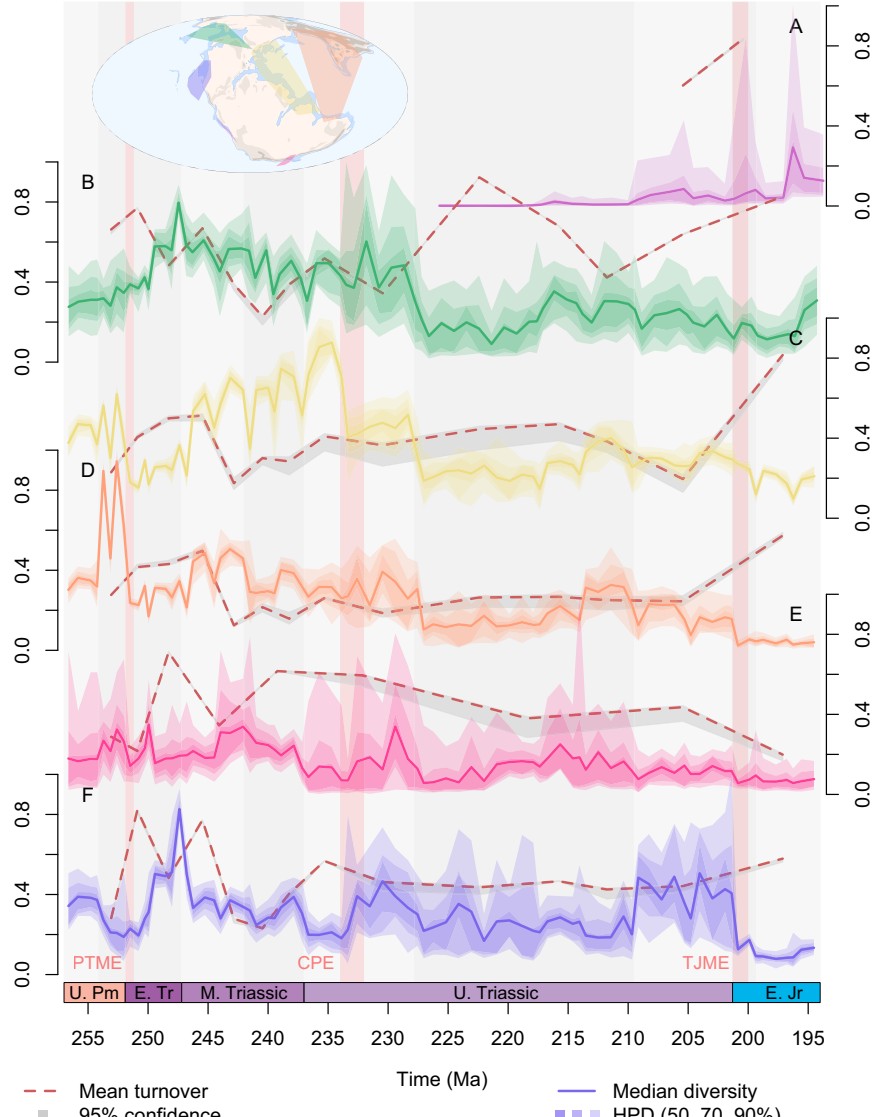

**Fig. 4 Regional probabilistic diversity curves, their 50%, 75% and 95% highest posterior densities (HPD), and median turnover with 95% confidence intervals.** Diversity trajectories and magnitude of turnover, differ substantially between different parts of the globe and across key biotic upheavals highlighted with red bars: the end-Permian mass extinction (PTME), Carnian Pluvial Episode (CPE), and end-Triassic mass extinction (TJME). Full uncertainty intervals for the diversity estimates are displayed here, but show that the median diversity trends presented in Fig. 2 are representative of these uncertainties. As in Fig. 2, displayed results are derived from datasets standardised by minimum spanning tree length. All diversity values are rescaled so that the highest value in each 95% confidence interval is equal to 1. Each turnover value and confidence interval record the relative abundance-corrected Forbes* dissimilarity relative to the preceding bin. **A** South Panthalassic. **B** Boreal. **C** West Circumtethys. **D** East Circumtethys. **E** Tangaroan. **F** North Panthalassic. Source data are provided as a Source Data file.

Circumtethys comprise the largest portion of the composite dataset and the region shows a similar taxonomic composition to that of the total dataset (Fig. 7). This is not unexpected given the historical intensity of sampling in Europe[9] but suggests that the data from West Circumtethys exert a disproportionate influence on global diversity trends at least for our study interval. The regional heterogeneity we recover further demonstrates that the quasi 'global' signal from West Circumtethys is not representative of diversity dynamics elsewhere. Consequently, major biotic events described from the supposedly global fossil record must be scrutinised to determine the degree to which they manifest at a regional level, or whether they are primarily West Tethyan phenomena.

Regional diversity dynamics all support the PTME as a global event (Figs. 3 and 4), but extinction intensity shows a degree of latitudinal structuring between regions. The greatest deficits in origination rates and diversity crashes occur in the Circumtethyan and North Panthalassic regions (Fig. 5C, D, F), which strongly sample the equator and tropics, in contrast to more modest deficits and declines in the high latitude Boreal and Tangaroan regions. This is consistent with the geological evidence for extreme ocean warming at low latitudes[35], along with the flattening of the latitudinal diversity gradient across the equator and tropics in the earliest Triassic[36]. Recovery from the PTME is also regionally heterogenous. Extinction rates remain high throughout the earliest Triassic, but soon dip below a relatively constant origination rate in the high latitude Tangaroan and Boreal regions (Fig. 5B, E). The credible intervals on origination rates in the latter, however, indicate that spikes of origination may have taken place in the earliest Triassic, indicating that the PTME and its aftermath may have

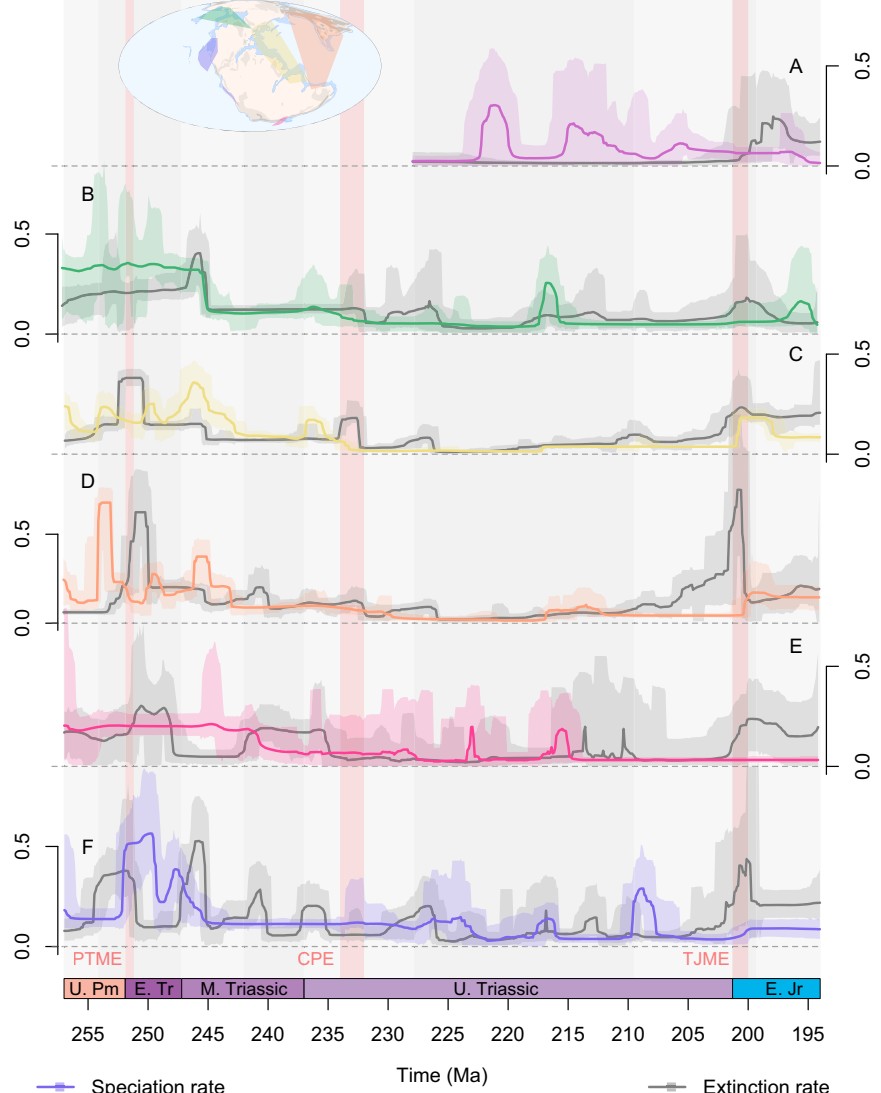

**Fig. 5 Probabilistic speciation and extinction rates and their 95% highest posterior densities through the Late Permian to Early Jurassic (259.5–192.9 Ma).** Extinction and speciation rates differ markedly between sampling regions through our study interval, even during globally pervasive biological upheavals like the end-Permian mass extinction (PTME), Carnian Pluvial Episode (CPE), or end-Triassic mass extinction (TJME). Deconvolving the diversification rates in Fig. 3 into the component speciation and extinction processes is additionally important to capture pulses of cryptic turnover where net diversification and so the change in diversity is only small. Displayed results are derived from PyRate applied to datasets standardised by minimum spanning tree length, meaning that the results are unbiased by heterogenous sampling through time and across space. **A** South Panthalassic. **B** Boreal. **C** West Circumtethys. **D** East Circumtethys. **E** Tangaroan. **F** North Panthalassic. Source data are provided as a Source Data file.

manifested as pulses of turnover rather than a steady increase in diversity. Steady recovery is instead seen in the Circumtethyan regions, with modest spikes in median origination rate in the wake of the extinction pulse (Fig. 5C, D). In the North Panthalassic region, however, massive spikes in origination far in excess of extinction take place in the immediate aftermath of the PTME. Although this pattern may be influenced by the change in the spatial extent of the data, the confidence interval on extinction rate still clearly picks out the PTME, while the peaks in origination rate fall fully within the well-standardised portion of the dataset (Fig. 5F). This confirms rapid and strong recovery from the event in this region and is well supported by the existence of widespread and exceptionally diverse marine assemblages just three million years after the PTME in the North Panthalassic region[37,38]. These differences may indicate different ecological dynamics underpinning the recovery at different latitudes, with re-entry of surviving or opportunistic lineages into newly vacated niches at low latitudes

versus chaotic patterns of turnover at high latitudes, driven by the invasion of survivors in ecologically stressed refugia[36].

The timing and placement of pulses of origination and extinction throughout the Middle Triassic are variable and do not correspond to any proposed global events. This heterogeneity continues through the Carnian and may reflect the role of regionally unique macroecological influences on diversity along with the regionally variable quality of the fossil record. Sedimentological evidence for regionally synchronous environmental upheaval during the CPE is globally pervasive[28,39,40] and four distinct pulses of volcanism and carbon isotope excursion, linked to the eruption of the Wrangellia Large Igneous Province, can be identified with confidence during the CPE in both East[41] and West Circumtethys[42]. Only West Circumtethys, however, shows the signal of biotic crisis during the CPE, with strongly negative diversification rates and a sharp crash in diversity (Fig. 3C). A diversity crash is also well supported in the Tangaroan region, but

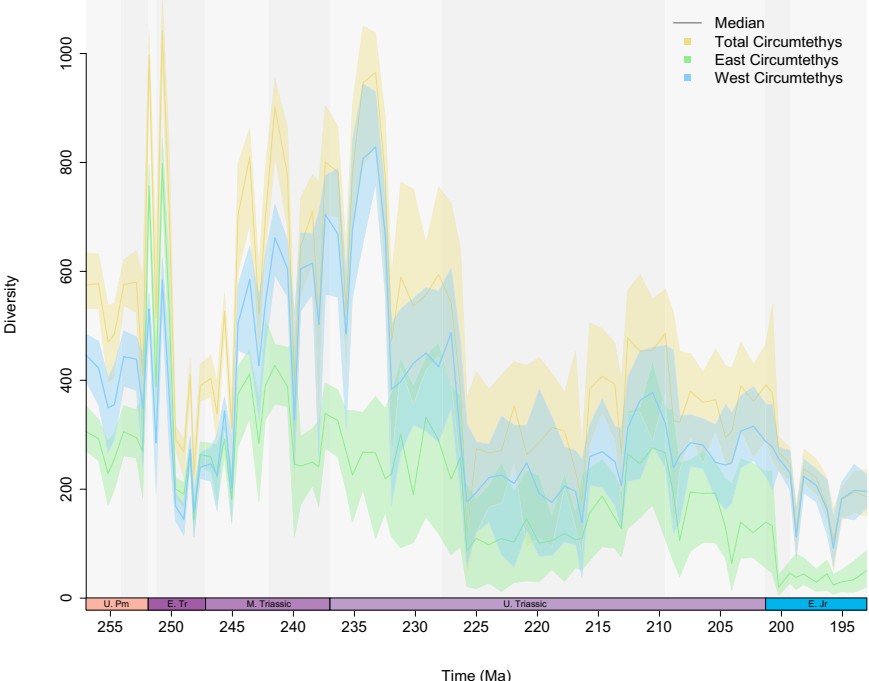

**Fig. 6 Probabilistic median diversity and 50%, 75% and 95% highest posterior densities.** Diversity trajectories derived from the total Circumtethys region and its eastern and western subdomains, highlighting the dissonance in trends between regions and at different spatial scales. Diversity dynamics at broader spatial scales are therefore not simply the result of upscaling of regional processes. The greater degree of congruence between the total Circumtethyan dataset and its larger subcomponent (West Circumtethys) additionally demonstrates how oversampling of one particular region may end up driving the global trend. Displayed results are derived from MST-standardised datasets. Source data are provided as a Source Data file.

the diversification rates show high uncertainty (Fig. 3E), while negative diversification is present in the Boreal and East Circumtethys regions but without a substantial crash in diversity (Fig. 3B, D). Conversely, diversity increases sharply in the North Panthalassic region with an accompanying pulse of strong diversification (Fig. 3F). Intriguingly, there is more consistent evidence in each region for a diversity crash at the Carnian-Norian boundary in all regions, bar the South Panthalassic which does not extend to this interval. While there is some geological evidence in East Circumtethys for genuine environmental fluctuations at the end of the Carnian[43], it may instead be the case that the temporal resolution of many of the occurrences in each region is driving this signal. Even though most of our data is constrained to substage level, for stages divided into an early and a late substage (as is the case for the Carnian), FADs of early substage occurrences and LADs of late substage occurrences will still coincide with stage-level divisions and so may continue to drive apparent changes in rates and diversity across these boundaries. This suggests that the Permo-Jurassic data in the PBDB may be approaching its analytical limit, even when coupled with model-based estimation methods that can account for temporal uncertainty. There is no strong change in turnover in any region across the CPE or the Carnian-Norian boundary. While there is still dissimilarity ranging from 0.2 to 0.5, there are no sharp increases in turnover that would otherwise be expected as a result of a sudden crash in diversity. Consequently, the ecological signature of turnover throughout the Carnian appears subdued compared to that across the PTME.

Compared to the PTME, the signal of the TJME is more complex. The onset of negative diversification rates at the TJ boundary is abrupt in all regions, aside from East Circumtethys where they become steadily more negative throughout the Rhaetian (Fig. 3D) and with only weak support in the Tangaroan. Given our mechanism of FAD-LAD sampling, the sharp contraction in spatial

extent we noted during our standardisation protocol is expected to mute origination and extinction rates during the Hettangian, suggesting that the strongly negative diversification signal is genuine. Diversity loss around the TJ boundary is only substantial in the North Panthalassic region (Fig. 3F) but reduced in the others, further indicating that it is a poor proxy for diversification dynamics. In the Boreal and West Circumtethys regions, turnover shows only a modest increase across the TJ boundary (Fig. 4B, C), following on from steadily increasing dissimilarity throughout the Late Triassic, suggesting that the ecological impact of the event merely represented the zenith of long-term turnover starting well before the extinction boundary. In the Tangaroan region, however, turnover declines across the event (Fig. 3E), showing that the change in faunal composition of the region across the extinction boundary was not as marked compared to earlier change taking place throughout the Late Triassic.

High-resolution records of the TJME from stratigraphic sections confirm that the event was complex, with multiple pulses of extinction separated by a few hundred thousand years[44], and mercury anomalies indicating that continued eruptive phases of the Central Atlantic Magmatic Province (CAMP) and hostile environmental conditions extended into the Hettangian by a similar degree[45–47], matched by the persistent negative diversification rates in each region throughout the Hettangian. This is therefore unusual given the more muted changes in diversity across the event. Analysis of the Phanerozoic fossil record as a series of eco-evolutionary units based on taxon co-occurrences through time has shown that the TJME had a significant impact at the ordinal level, with prominent ecological restructuring particularly among reef communities, but little impact at the family or generic levels[48]. Thus, while strong ecological and environmental change certainly took place at the TJ boundary in concert with CAMP volcanism[49], this may have been predicated on relatively small generic changes suggestive of the loss of

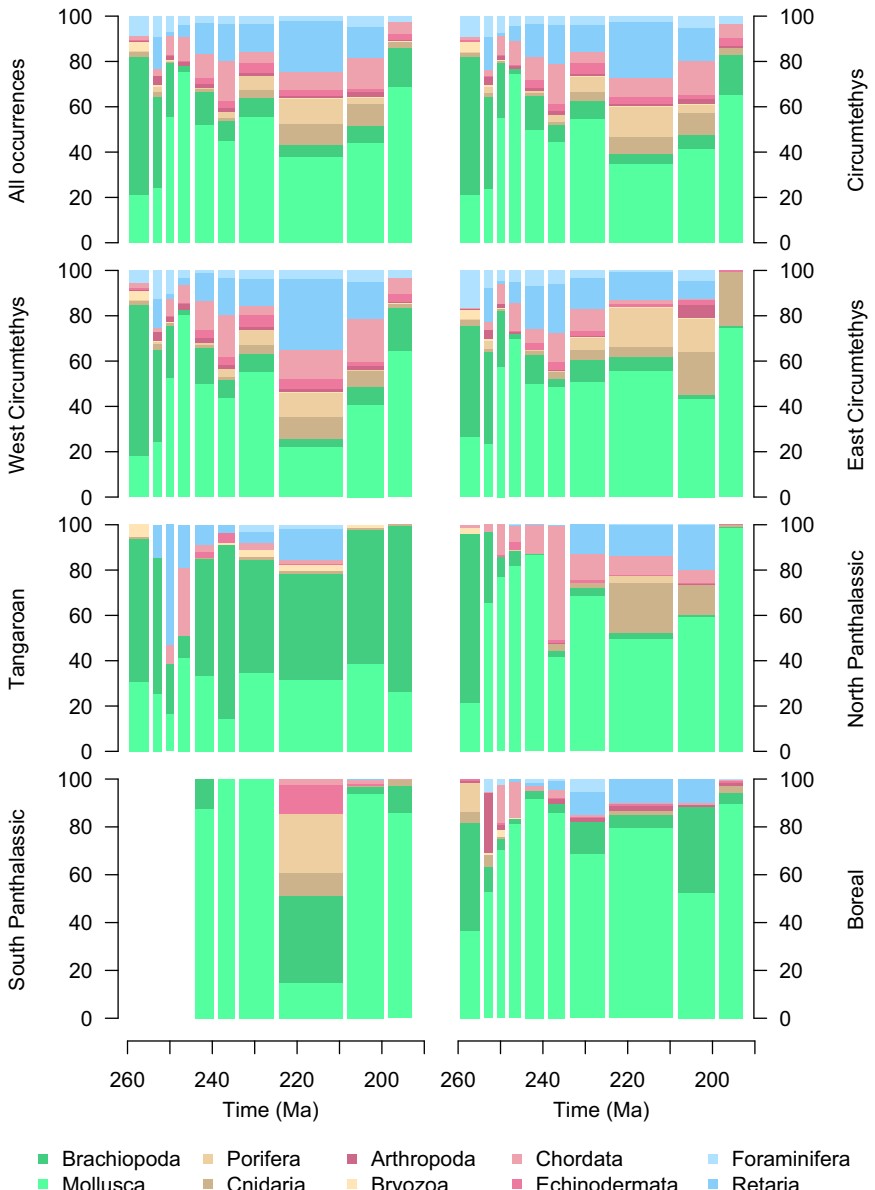

**Fig. 7 Taxonomic compositions of the total dataset and each sampling region through the Late Permian to Early Jurassic (259.5–192.9 Ma).**
Comparison of the total dataset to the West Circumtethys dataset indicates the dominance of the latter within our analyses (see Fig. 5 also). The prevalence of cephalopods also indicates the potential for taphonomic bias with regard to the taxonomic affiliations of the fossil occurrences, indicating that regional and even global views of diversity dynamics through our study interval will largely reflect the tempo of one or a handful of different clades. Source data are provided as a Source Data file.

keystone species. Dunhill et al.[50] similarly noted little change in generic or functional richness at the TJ boundary when analysing PBDB data with traditional bin-based approaches, and also found a reduced impact of the TJME in the Tethys and Boreal oceans compared to the Panthalassic, supporting those aspects of our results and further suggesting that the ecological and taxonomic severities of the TJME are somewhat decoupled.

There is strong correspondence between global diversity in deep time and the history of reef ecosystems[48,51], with reefs acting as cradles of biodiversity and evolutionary innovation throughout the Phanerozoic[52] but displaying high sensitivity to strong environmental disturbances such as those during mass extinctions[53]. We tentatively identify two key instances of this relationship from our analyses. The strongest evidence for biotic upheaval during the CPE comes from West Circumtethys (Fig. 3C), driven by the decline of carbonate platforms and hyper-

diverse reef assemblages in the European geological record[54]. On this basis, it has been proposed previously that not only is the CPE a primarily West Tethyan phenomenon, but also that the apparent scale of the diversity crash is exacerbated by the loss of these assemblages and environments[55,56]. The evidence for environmental perturbation during the CPE is globally distributed[28], however, and there is evidence for diversity decline in other regions to some extent. In a global diversity curve, the loss of ecologically diverse West Tethyan reef systems may be viewed as a statistical artefact, but our decomposition of the global signal into regional subsets transforms this artefact into an empirical aspect of Tethyan biogeographic history. As a modern analogue, the Great Barrier Reef is individually one of the most diverse habitats on the planet and its decline is viewed as a genuine and catastrophic aspect of the current global diversity crisis[57], rather than as a regional anomaly.

A similar pattern is present in East Circumtethys during the Late Permian where the development of ecologically diverse reef systems across a regionally extensive carbonate platform[58] coincides with a large pulse of origination and a corresponding diversity zenith, followed by catastrophic extinction and diversity loss at the PTME (Fig. 3D). Our regional analyses highlight the spatial component of the correspondence between reef systems and Phanerozoic marine diversity, with the regional loss of reef systems contributing substantially to global marine diversity crises. Thus, while large evolutionary events may have global signatures in the fossil record, they may also display regional epicentres due to the interactions of spatially non-random controls on diversity with diversity drivers operating at global scales. Across the TJME, reefs were widely distributed, and so their relationship with global diversity approached a global trend, rather than displaying any distinct regionalisation, with previous studies confirming the severity of the event for reefs globally[50,59,60].

Our approach to examining the fossil record provides a powerful way to decompose global diversity trends into their regional components, but the scope of the approach remains reliant on the availability of high-quality occurrence data. As such, we believe that our methods will be well suited to examining major biotic events in other transects of geological history, for example the poorly resolved Late Devonian mass extinction. Full resolution of some events, however, may be hindered by the current quality of fossil occurrence data. Continued analytical gain will come from refinement of occurrence ages, either through the literature-based approach applied here or through stratigraphic modelling approaches like CONOP.SAGA[61]. Similarly, our regional view of Triassic diversity dynamics will be aided by improved spatial coverage of the fossil record, although this remains contingent on the availability of fossiliferous sedimentary rocks around the globe. Otherwise, a nuanced understanding of the differences between diversification signals at the section level will continue to provide a fine-controlled means of decomposing global biotic history into its regional components.

## Methods

**Spatial standardisation workflow.** To produce spatially-standardised fossil occurrence datasets which remain geographically consistent through time, we designed a subsampling algorithm which enforces consistent spatial distribution of occurrence data between time bins, while maximising data retention and permitting highly flexible regionalisation (Fig. 8). Our method was developed in light of, and takes some inspiration from, the spatial standardisation procedure of Close et al.[2]. This method provides, within a given time bin, subsamples of occurrence data with threshold MST lengths. An average diversity estimate can be taken from this 'forest' of MSTs, selecting only those of a target tree length to ensure spatially-standardised measurements. It does not produce a single dataset across time bins, however; rather a series of discontinuous, bin-specific datasets which cannot then simply be concatenated as the spatial extents of each bin-specific forest are not standardised (despite each individual MST being so), even when MSTs are assigned to a specific geographic region, e.g. a continent or to a particular latitudinal band. This prevents estimation of rates, because such analyses require datasets that span multiple time bins and remain geographically consistent and spatially standardised through the time span of interest. This is the shortcoming that our method overcomes. The workflow consists of three main steps.

1. First, the user demarcates a spatially discrete geographic area (herein the spatial window) and a series of time bins into which fossil occurrence data is subdivided. Occurrence data falling outside the window in each time bin are dropped from the dataset, leaving a spatially restricted subsample (Fig. 8A). Spatial polygon demarcation is a compromise between the spatial availability of data to subsample and the region of interest to the user but allows creation of a dataset where regional nuances of biodiversity may be targeted. Careful choice of window extent can even aid subsequent steps by targeting regions that have a consistently sampled fossil record through time, even if the extent of that record fluctuates. To account for spatially non-random changes in the spatial distribution of occurrence data arising from the interlinked effects of continental drift, preservation potential and habitat distribution[17], the spatial polygon may slide to track the location of the available sampling data through time. This drift is performed with two conditions. First, the drift is unidirectional so that the sampling of data remains consistent relative to global geography, rather than allowing the window to hop across the globe solely according to data availability and without biogeographic context. Second, spatial window translation is performed in projected coordinates so that its sampling area remains near constant between time bins, avoiding changes in spatial window area that could induce sampling bias from the species-area effect.

2. Next, subsampling routines are applied to the data to standardise its spatial extent to a common threshold across all time bins using two metrics: the length of the MST required to connect the locations of the occurrences; and the longitude–latitude extent of the occurrences. MST length has been shown to measure spatial sampling robustly as it captures not just the absolute extent of the data but also the intervening density of points, and so is highly correlated with multiple other geographic metrics[16]. MSTs with different aspect ratios may show similar total lengths but could sample over very different spatial extents, inducing a bias by uneven sampling across spatially organised diversity gradients[16]; standardising longitude–latitude extent accounts for this possibility. The standardisation methods can be applied individually or serially if both MST length and longitude–latitude range show substantial fluctuations through time. Data loss is inevitable during subsampling and may risk degrading the signals of origination, extinction and preservation. To address this issue, subsampling is performed to retain the greatest amount of data possible. During longitude–latitude standardisation, the range containing the greatest amount of data is preserved. During MST standardisation, occurrences are spatially binned using a hexagonal grid to reduce computational burden and to permit assessment of spatial density (Fig. 8B). The grid cells containing the occurrences that define the longitude–latitude extent of the data are first masked from the subsampling procedure so that this property of the dataset is unaffected, and then the occurrences within the grid cells at the tips of the MST are tabulated. Tip cells with the least data are iteratively removed (removal of non-tip cells may have little to no effect on the tree topology) until the target MST length is achieved (Fig. 8D), with tree length iteratively re-calculated to include the branch lengths added by the masked grid cells.

For both methods, the solution with the smallest difference to the target is selected and so both metrics may fluctuate around this target from bin to bin, with the degree of fluctuation depending upon the availability of data to exclude—larger regions that capture more data are more amenable to the procedure than smaller regions. Similarly, the serial application of both metrics reduces the pool of data available to the second method, although longitude–latitude standardisation is always applied first in the serial case so that the resultant extent will be retained during MST standardisation. Consequently, the choice of standardisation procedure and thresholds must be tailored to the availability and extent of data within the sampling region through time, along with the resulting degree of data loss. This places further emphasis on the careful construction of the spatial window in the first step. Threshold choice is also a compromise between data loss and consistency of standardisation across the dataset and so it may be necessary to choose targets that standardise spatial extent well for the majority of the temporal range of a dataset, rather than imposing a threshold that spans the entire data range but causes unacceptable data loss in some bins.

3. Once the time-binned, geographically restricted data have been spatially standardised, the relationship between diversity and spatial extent is scrutinised. After standardisation, it is expected that residual fluctuations in spatial extent should induce little or no change in apparent diversity. Bias arising from temporal variation in sampling intensity may still be present, so diversity is calculated using coverage-based rarefaction (also referred to as shareholder quorum subsampling[13,62,63]), with a consistent coverage quorum from bin to bin. While coverage-based rarefaction has known biases, it remains the most accurate non-probabilistic means of estimating fossil diversity[14]. As such, we consider it the most appropriate method to assess the diversity of a region-level fossil dataset. The residual fluctuations in spatial extent may then be tested for correlation with spatially standardised, temporally corrected diversity. If a significant relationship is found, then the user must go back and alter the standardisation parameters, including the spatial window geometry and drift, the longitude–latitude threshold, and the MST threshold. Otherwise, the dataset is considered suitable for further analysis.

We implement our subsample standardisation workflow in R with a custom algorithm, *spacetimestand*, along with a helper function *spacetimewind* to aid the initial construction of spatial window. *spacetimestand* can then accept any fossil occurrence data with temporal constraints in millions of years before present and longitude–latitude coordinates in decimal degrees. Spatial polygon construction and binning is handled using the sp library[64], MST manipulation using the igraph and ape libraries[65,66], spatial metric calculation using the sp, geosphere and GeoRange libraries[67,68], hexagonal gridding using the icosa library[69], and diversity calculation by coverage-based rarefaction using the *estimateD* function from the iNEXT library[70]. Next, we apply our algorithm to marine fossil occurrence data from the Late Permian to Early Triassic.

**Data acquisition and cleaning.** Fossil occurrence data for the Late Permian (260 Ma) to Early Jurassic (190 Ma) were downloaded from the PBDB on 28/04/21 with the default major overlap setting applied (an occurrence is treated as within the requested time span if 50% or more of its stratigraphic duration intersects with that time span), in order to minimise edge effects resulting from incomplete sampling of taxon ranges within our study interval of interest (the Permo-Triassic

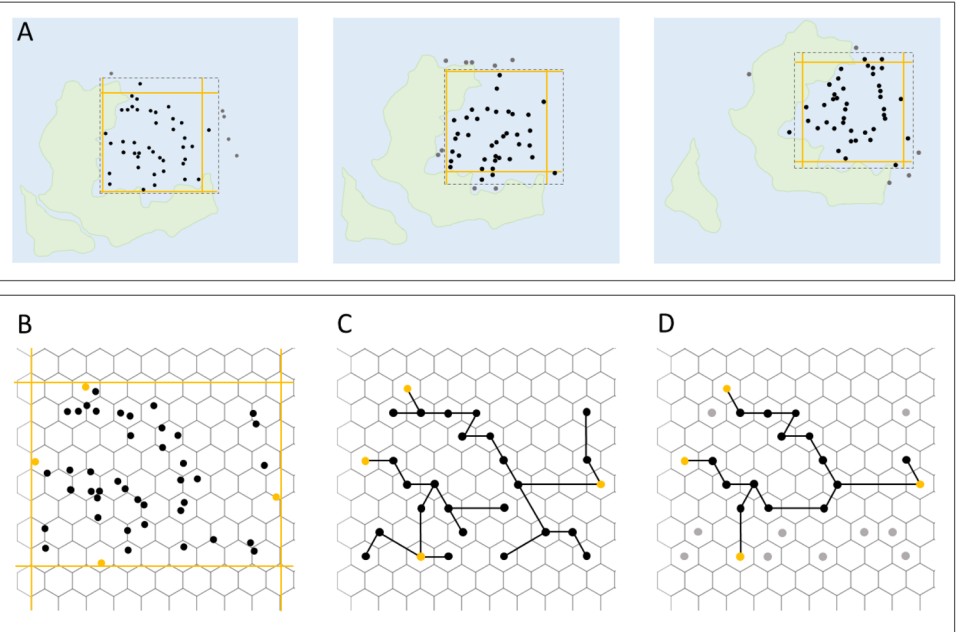

**Fig. 8 Component steps of our spatial standardisation workflow. A** A spatial window (dotted lines) is used to demarcate the spatial region of interest, which may shift in a regular fashion through time to track that region. Data captured in each window is clipped to a target longitude–latitude range (orange lines). **B** The data forming the longitude–latitude extent is marked, then masked from further subsampling. **C** Data are binned using a hexagonal grid, the tally of occurrences in each grid cell taken, and a minimum spanning tree constructed from the grid cell centres. **D** The cells with the smallest amount of data are iteratively removed from the minimum spanning tree until a target tree length is reached.

to Triassic-Jurassic boundaries). Other filters in the PBDB API were not applied during data download to minimise the risk of data exclusion. Occurrences from terrestrial facies were excluded, along with plant, terrestrial-freshwater invertebrate and terrestrial tetrapod occurrences (as these may still occur in marine deposits from transport) and occurrences from several minor and poorly represented phyla. Finally, non-genus level occurrences were removed, leaving 104,741 occurrences out of the original 168,124. Based on previous findings[2], siliceous occurrences were not removed from the dataset, despite their variable preservation potential compared to calcareous fossils. To increase the temporal precision of the dataset, occurrences with stratigraphic information present were revised to substage- or stage-level precision using a stratigraphic database compiled from the primary literature. To increase the spatial and taxonomic coverage of the dataset, the PBDB data were supplemented by an independently compiled genus-level database of Late Permian to Late Triassic marine fossil occurrences[36]. Prior to merging, occurrences from the same minor phyla were excluded, along with a small number lacking modern coordinate data, leaving 47,661 occurrences out of an original 51,054. Absolute numerical first appearance and last appearance data (FADs and LADs) were then assigned to the occurrences from their first and last stratigraphic intervals, based on the ages given in A Geologic Timescale 2020[71]. Palaeocoordinates were calculated from the occurrence modern-day coordinates and midpoint ages using the Getech plate rotation model. Finally, occurrences with a temporal uncertainty greater than 10 million years and occurrences for which palaeo-coordinate reconstruction was not possible were removed from the composite dataset, leaving 145,701 occurrences out of the original 152,402.

In the total dataset, we note that the age uncertainty for occurrences is typically well below their parent stage duration, aside for the Wuchiapingian and Rhaetian where the mean and quartile ages are effectively the same as the stage length (Fig. S44). This highlights the chronostratigraphic quality of our composite dataset, particularly for the Norian stage (~18-million-year duration) which has traditionally been an extremely coarse and poorly resolved interval in Triassic-aged macroevolutionary analyses. Taxonomically, most occurrences are molluscs (Fig. 8), which is unsurprising given the abundance of ammonites, gastropods and bivalves in the PBDB, but introduces the caveat that downstream results will be driven primarily by these clades. Foraminiferal and radiolarian occurrences together comprise the next most abundant element of the composite dataset, demonstrating that we nonetheless achieve good coverage of both the macrofossil and microfossil records, along with broad taxonomic coverage in the former despite the preponderance of molluscs.

**Spatiotemporal standardisation.** We chose a largely stage-level binning scheme when applying our spatial standardisation procedure for several reasons. First, the volume of data in each bin is greater than in a substage bin, providing a more stable view of occurrence distributions through time and increasing the availability of

data for subsampling. Spatial variation at substage level might still affect the sampling of diversity, but the main goal of this study is to analyse origination and extinction rates where taxonomic ranges are key rather than pointwise taxonomic observations. Consequently, substage level variation in taxon presences likely amounts to noise when examining taxonomic ranges, making stage-level bins preferable in order maximise signal.

During exploratory standardisation trials, we found a large crash in diversity and spatial sampling extent during the Hettangian (201.3–199.3 Mya). No significant relationships with spatially-standardised diversity were found when the Hettangian bin was excluded from correlation tests, indicating its disproportionate effect in otherwise well-standardised time series. Standardising the data to the level present in the Hettangian would have resulted in unacceptable data loss so we instead accounted for this issue by merging the Hettangian bin with the succeeding Sinemurian bin, where sampling returns to spatial extents consistent with older intervals. While this highlights a limitation of our method, as the Hettangian is <2 Ma in length, it is reasonable to expect it would have a minor effect on taxonomic ranges in the long term, despite the magnitude of the sampling crash, and that any taxa surviving through the interval will be recorded in the much better sampled Sinemurian.

The occurrence data were plotted onto palaeogeographic maps to identify biogeographic regions that could feasibly be subsampled consistently through time. We identified five such regions which broadly correspond to major Permo-Jurassic seaboards and ocean basins: Circumtethys, Boreal, North Panthalassic and South Panthalassic, along with an unexpected set of marine occurrences from the Australian and New Zealand fossil record, which we term the *Tangaroan* (so named for the Maori god of the oceans). As Circumtethys is an extremely large region compared to the others, we subdivide it into eastern and western subdomains. While the extent of spatial regions reflects a compromise between biogeographic discretion and data availability and can theoretically be arbitrary, we note that most of our regions share a degree of correspondence with bioregions for the Permo-Triassic predicted from abiotic drivers of marine provinciality[72], suggesting that they are biologically realistic to a certain extent. The major exception to this is our east-west division of Tethys compared to the north-south divide recovered by Kocsis et al.[32,33,72] as this was a compromise between biogeographic realism and data availability through time.

All regions extend for the full temporal range of the composite dataset, aside from the South Panthalassic, which covers the Late Triassic to Early Jurassic. Spatial windows were constructed for each region using the *spacetimewind* R function, then data were subsampled into each region under the described binning strategy using the *spacetimestand* R function. Four treatments were conducted for each polygon-binned dataset: no standardisation, standardisation by MST length, standardisation of the longitude–latitude extent and standardisation with both methods. For each treatment in each region, bin-wise diversity was calculated using coverage-based rarefaction at coverage levels of 40, 50, 60 and 70% (Figs. S8–S14).

The relationships between diversity at each level of coverage with MST length, longitude range and latitude range were interrogated using one-tailed Pearson's product moment and Spearman's rho tests of correlation, with Benjamini–Hochberg correction for multiple comparisons[73]. Spatial standardisation protocols for each region were then adjusted to eliminate significant correlations as needed.

**Rate data and preservation model**. Origination, extinction, and preservation rates were jointly estimated in a Bayesian framework using PyRate (v3.0). PyRate implements realistic preservation models that can vary through time and among taxa, yielding substantial increases in rate estimation accuracy over traditional methods. The method can also model occurrence-age uncertainty and provides an explicit model-based means of testing whether proposed rate shifts are significant[23]. A comparable approach is the FBD-range model which accounts for unsampled diversity, something PyRate cannot do by default, but assumes an unrealistic constant preservation rate[74]. An implementation of FBD-range is present experimentally within PyRate, but the complexity of the analysis currently renders this method computationally intractable for large datasets. Regardless, the FBD-range model and PyRate have been compared against one another, as well as against results from traditional methods, with FBD and PyRate showing largely comparable performance (although FBD remains more accurate under some scenarios of lower preservation rates and high turnover) and both FBD and PyRate outstrip traditional methods significantly[74].

PyRate has been criticised recently for only performing well when data availability is high and consistently sampled[75]. This criticism, however, was based on simulated data with an underlying phylogenetic structure parameterised from a tree of ornithischian dinosaurs, whose fossil record is known to be inconsistent[76] and is at odds with the findings of simulations covering a broader range of turnover and preservation rates[74]. PyRate is demonstrably subject to the pitfall of spatial variability in the fossil record, with regional analyses of the crocodylomorph fossil record indicating declining diversity[77], while global analysis with PyRate spuriously recovers increasing diversity driven by expansion of the geographic range of their fossil record[17,77,78]. We avoid the issue of spatial variability with our standardisation procedure and the marine fossil record is well-sampled compared to the scenarios where PyRate otherwise begins to perform poorly. Therefore, we assert that PyRate is a suitable method for inferring diversity dynamics from our dataset and we elect not to use traditional methods (e.g. boundary crossers or three-timer rates[79]).

We analysed datasets from the unstandardised, MST-standardised and MST + longitude–latitude-standardised treatments; as MST length is the most important control on spatial extent, the dataset with longitude–latitude standardisation only is expected to retain significant spatial bias. Ten age-randomised input datasets for each region and data treatment were generated in R with locality-age dependence (all occurrences from a locality are given the same randomised age), using collection number as a proxy for locality for PBDB-derived occurrences and geological section names for occurrences from the independent dataset. Locality-age dependence is both logically desirable as locality occurrences strictly represent a geographically localised and temporally discrete fauna (in idealised terms an assemblage from a single bedding plane) and which has been shown to improve precision in age estimates in other Bayesian dating procedures using fossil data[80].

The best fitting preservation model (homogenous, HPP; non-homogenous, NHPP; or time-variable homogenous Poisson process TPP) for each dataset was identified by maximum likelihood using the -PPmodeltest function of PyRate, with the best fitting model identified using the Akaike Information Criterion[81]. In addition to testing between the HPP, NHPP and TPP preservation models, we also tested between three TPP models of differing complexity: one with stage-level bins, one with stage-level bins and subdivision of the Norian stage into three sub-bins (the informal divisions Lacian, Alaunian and Sevatian), and one with substage-level bins and subdivision of the Norian stage into three sub-bins. For all datasets, the last binning scheme was found to be the best fitting, despite the greater number of model parameters (individual time-bin preservation rates) that it introduces. As well as using the TPP model of preservation through time with substage-level bins and threefold subdivision of the Norian, the preservation rate was also allowed to vary according to a gamma distribution (here discretized into eight rate multipliers[22,82]) on taxon-wise preservation rates. While there is currently no way to test between preservation models with and without the gamma parameter in PyRate, it is a recommended addition due to the known empirical variability of preservation rates among taxa, especially for taxonomically diverse datasets and because it includes a single additional parameter in the model. In each analysis, the bin-wise preservation rates were assigned a gamma prior with fixed shape parameter set to 2, while the scale parameter was itself assigned a vague exponential hyperprior and estimated through MCMC (PyRate option -pP 2 0). This hierarchical approach provides a means of regularisation while allowing the prior on the preservation to adapt to the dataset[23]. Finally, rate shifts outside the covered range of the data were excluded in each analysis to avoid edge effects during parameter estimates (PyRate option -edgeShift).

**Rate estimation**. Regardless of the chosen preservation model, a PyRate analysis is parameter-rich as the individual origination and extinction times for each taxon are jointly estimated along with the overall rates. PyRate additionally uses a

reversible-jump Markov Chain Monte Carlo (rjMCMC) with a standard Metropolis Hastings algorithm to sample parameters across models with different numbers of rate shifts. This produces high computational burden, and models for larger sampling regions could not be estimated efficiently. PyRate can alternatively use an efficient Gibbs algorithm to sample from the posterior distribution of the parameters, producing preservation-corrected estimates of origination and extinction times that are virtually identical to those from the Metropolis Hastings algorithm, but with a coarse birth-death model that involves a dramatic loss of resolution in the resulting rate curves[83]. A second programme, LiteRate, has been developed to permit origination and extinction rate estimation for taxonomically large datasets[24,25], gaining computational efficiency by implementing the same birth-death model used by PyRate with the rjMCMC and Metropolis Hastings algorithm, but without estimation of the complex preservation model. As we expect ranges in a fossil dataset to be truncated by variation in preservation rate through time, times of origination and extinction would be inaccurately estimated if LiteRate were run directly with a fossil dataset.

To overcome these methodological issues, we use a two-step procedure to permit efficient model estimation for taxonomically large fossil datasets. First, we use PyRate with the Gibbs algorithm to jointly estimate the parameters of the preservation model and the preservation rate-corrected estimates of origination and extinction times for each taxon. The origination and extinction time estimates are then supplied as input in LiteRate, leaving only the estimation of rates and rate shifts from the computationally efficient birth-death model. In summary, PyRate is used to perform the computationally expensive task of estimating the complex preservation model parameters and taxon-specific origination and extinction times using the computationally efficient Gibbs algorithm, while LiteRate is used to estimate the high-resolution birth-death model, rates and rate shifts for the taxonomically large dataset.

PyRate analyses for each region were run across sets of ten age-randomised replicates for five million generations, aside for the Tangaroan (10 million) and South Panthalassic (20 million), with sampling rates set to produce 10,000 samples of the posterior. Output datasets were assessed using Tracer (v1.7.1)[84] to determine suitable burn-in values by visually inspecting the MCMC trace, and to check for convergence by ensuring minimum effective sample sizes on all model parameters of 200 post burn-in for each analysis. Mean origination and extinction times were derived using the -ginput function of PyRate with a 10% burn-in, before being supplied to LiteRate. LiteRate analyses for each region were run across the 10 sets of mean origination and extinction times for 200 million generations, aside for the South Panthalassic (250 million). To incorporate age uncertainty into each analysis, logs from each age-randomised replicate were combined respectively for PyRate and LiteRate using the -combLog function of PyRate, taking 100 random samples from each log post 10% burn-in, to give 1000 samples of the posterior across all age-randomised replicated. Rates were then plotted at 0.1 million-year intervals and statistical significance of rate shifts recovered by the rjMCMC assessed using Bayes factors (log BF > 2: positive support; log BF > 6: strong support)[85] using the -plotRJ function of PyRate.

**Probabilistic diversity estimation**. Traditional methods of estimating diversity do not directly address uneven sampling arising from variation in preservation, collection and description rates, and their effectiveness is highly dependent on the structure of the dataset. We present an alternative method to infer corrected diversity trajectories based on the sampled occurrences and on the preservation rates through time and across lineages as inferred by PyRate, which we term mcmcDivE. The method implements a hierarchical Bayesian model to estimate corrected diversity across arbitrarily defined time bins. The method estimates two classes of parameters: the number of unobserved species for each time bin and a parameter quantifying the volatility of the diversity trajectory.

We assume the sampled number of taxa (i.e. the number of fossil taxa, here indicated with $x_t$) in a time bin to be a random subset of an unknown total taxon pool, which we indicate with $D_t$. The goal of mcmcDivE is to estimate the true diversity trajectory $\mathbf{D} = \{D_1, D_2, \ldots, D_t\}$, of which the vector of sampled diversity $\mathbf{x} = \{x_1, x_2, \ldots, x_t\}$ is a subset. The sampled diversity is modelled as a random sample from a binomial distribution[86] with sampling probability $p_t$:

$$x_t \sim \text{Bin}(D_t, p_t) \qquad (1)$$

We obtain the sampling probability from the preservation rate ($q_t$) estimated in the initial PyRate analysis. If the PyRate model assumes no variation across lineages the sampling probability based on a Poisson process is $p_t = 1 - \exp(-q_t \times \delta_t)$, where $\delta_t$ is the duration of the time bin. When using a Gamma model in PyRate, however, the $q_t$ parameter represents the mean rate across lineages at time $t$ and the rate is heterogeneous across lineages based on a gamma distribution with shape and rate parameters equal to an estimated value $\alpha$.

To account for rate heterogeneity across lineages in mcmcDivE, we draw an arbitrarily large vector of gamma-distributed rate multipliers $g_1, \ldots, g_R \sim \Gamma(\alpha,\alpha)$ and compute the mean probability of sampling in a time bin as:

$$p_t = \frac{1}{R} \sum_{i=R} 1 - \exp(-q_t \times g_i \times \delta_t) \qquad (2)$$

We note that while $q_t$ quantifies the mean preservation rate in PyRate (i.e. averaged among taxa in a time bin $t$), the mean sampling probability $p_t$ will be

**13**

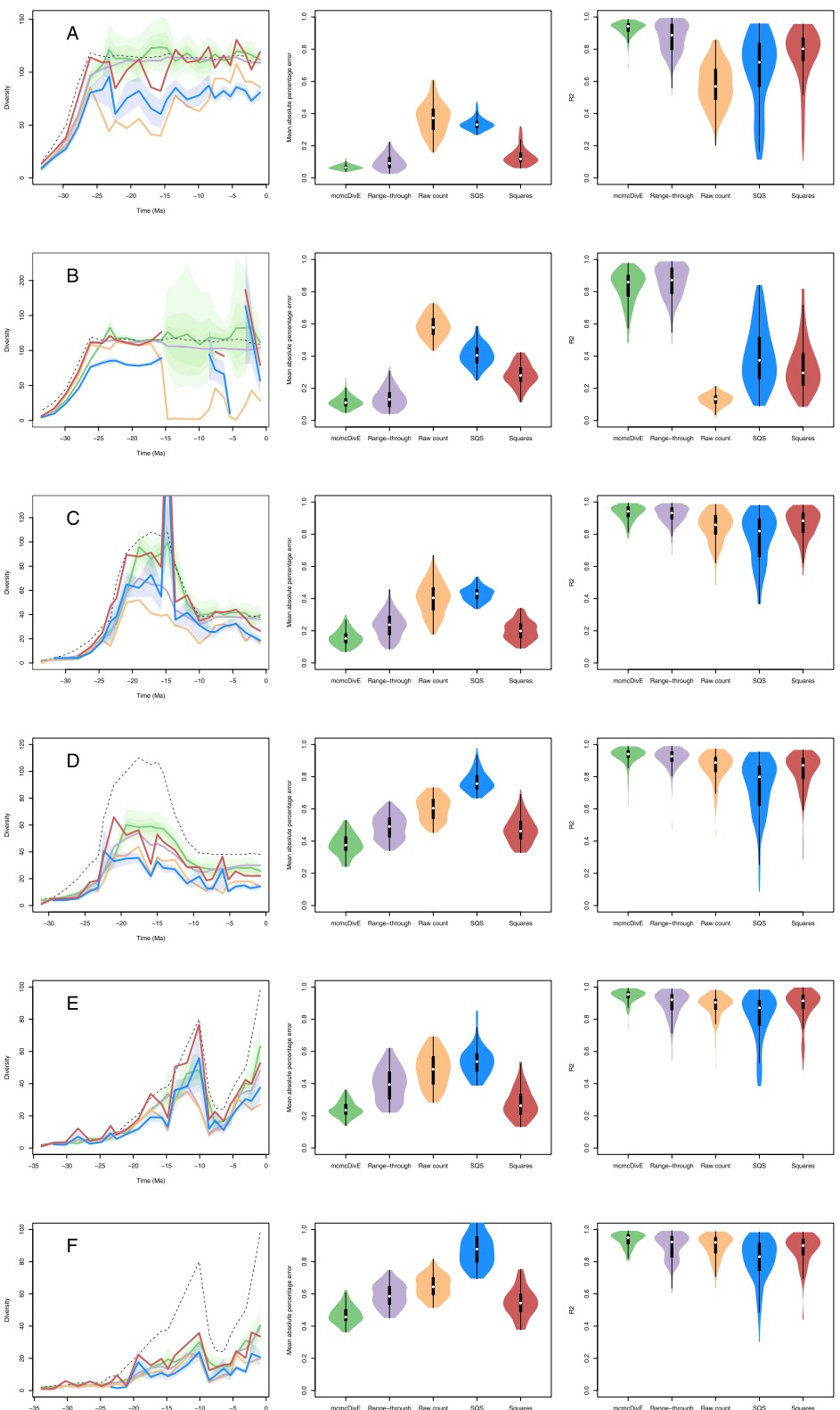

**Fig. 9 Validation of mcmcDivE accuracy using six different birth-death preservation simulations.** Plots in the left column show an example of diversity trajectories for one simulation: the true diversity is indicated with the dashed line, green line and shaded areas represent the mcmcDivE estimates and 95% confidence intervals. Purple, orange and red lines show the range-through, raw diversity and squares-extrapolated trajectories, respectively. Blue lines and shaded areas show diversity from coverage-based rarefaction/SQS and 95% confidence intervals. Violin plots show the distributions (range, 1st and 3rd quartiles and median) of absolute percentage errors and coefficient of determination between true and estimated diversity calculated from 100 simulations in each setting. The scenarios (**A**–**F**) refer to different birth-death and preservation settings as described in Table S65. Source data are provided as a Source Data file.

lower than $1 - \exp(-q_t \times \delta_t)$ (i.e. the probability expected under a constant preservation rate equal to $q_t$) especially for high levels of rate heterogeneity, due to the asymmetry of the gamma distribution and the non-linear relationship between rates and probabilities. We sample the corrected diversity from its posterior through MCMC. The likelihood of the sampled number of taxa is computed as the probability mass function of a binomial distribution with $D_i$ as the 'number of trials' and $p_i$ as the 'success probability'. To account for the expected temporal autocorrelation of a diversity trajectory[87], we use a Brownian process as a prior on the log-transformed diversity trajectory through time. Under this model, the prior probability of $D_t$ is:

$$P(\log(D_t)) \sim \mathcal{N}(\log(D_{t-1}), \sqrt{\sigma^2 \times \delta_t}) \tag{3}$$

where $\sigma^2$ is the variance of the Brownian process. For the first time bin in the series, $D_{t=0}$, we use a vague prior $\mathcal{U}(0, \infty)$. Because the variance of the process is itself unknown and may vary among clades as a function of their diversification history, we assign it an exponential hyperprior $\mathrm{Exp}(1)$ and estimate it using MCMC. Thus, the full posterior of the mcmcDivE model is:

$$\underbrace{P(D, \sigma^2 | x, q, \alpha)}_{\text{posterior}} \propto \underbrace{P(x | D, q, \alpha)}_{\text{likelihood}} \times \underbrace{P(D | \sigma^2)}_{\text{prior}} \times \underbrace{P(\sigma^2)}_{\text{hyperprior}} \tag{4}$$

where $\mathbf{D} = \{D_0, D_1, \ldots, D_t\}$ and $\mathbf{q} = \{q_0, q_1, \ldots, q_t\}$ are vectors of estimated diversity, sampled diversity, and preservation rates for each of $T$ time bins. We estimate the parameters $D$ and $\sigma^2$ using MCMC to obtain samples from their joint posterior distribution. To incorporate uncertainties in $q$ and $\alpha$ we randomly resample them during the MCMC from their posterior distributions obtained from PyRate analyses of the fossil occurrence data. While in mcmcDivE we use a posterior sample of $q_t$ and $\alpha$ precomputed in PyRate for computational tractability of the problem, a joint estimation of all PyRate and mcmcDivE parameters is in principle possible, particularly for smaller datasets. mcmcDivE is implemented in Python v.3 and is available as part of the PyRate software package.

**Simulated and empirical diversity analyses.** We assessed the performance of the mcmcDivE method using 600 simulated datasets obtained under different birth-death processes and preservation scenarios. The settings of the six simulations (A–F) are summarised in Table S65 and we simulated 100 datasets from each setting. Since the birth-death process is stochastic and can generate a wide range of outcomes, we only accepted simulations with 100 to 500 species, although the resulting number of sampled species decreased after simulating the preservation process. From each birth-death simulation we sampled fossil occurrences based on a heterogeneous preservation process. Each simulation included six different preservation rates which were drawn randomly within the boundaries 0.25 and 2.5, with rate shifts set to 23, 15, 8, 5.3 and 2.6 Ma. To ensure that most rates were small (i.e. reflecting poor sampling), we randomly sampled preservation rates as:

$$q \sim \exp(\mathcal{U}(\log(0.25), \log(2.5))) \tag{5}$$

In two of the five scenarios (D, F), we included strong rate heterogeneity across lineages (additionally to the rate variation through time), by assuming that preservation rates followed a gamma distribution with shape and rate parameters set to 0.5. This indicates that if the mean preservation rate in a time bin was 1, the preservation rate varied across lineages between <0.001 and 5 (95% interval). In one scenario (B), we set the preservation rate to 0 (complete gap in preservation) in addition to the temporal rate changes used in the other scenarios. Specifically, the preservation rate was set to 0 in two time intervals between 15 and 8 Ma and between 5.3 and 2.6 Ma.

We analyzed the occurrence data using PyRate to estimate preservation rates through time and infer the amount of rate heterogeneity across lineages. We ran 10 million MCMC generations using the TPP preservation rate model with gamma-distributed heterogeneity. We then ran mcmcDivE for 200,000 MCMC iterations assuming bins of 1-myr duration to estimate corrected diversity trajectories while resampling the posterior distributions of the preservation parameters inferred by PyRate. To summarise the performance of mcmcDivE we quantified the mean absolute percentage error computed as the absolute difference between true and estimated diversity averaged across all time bins and divided by the mean true diversity-through time, then used a one-tailed t-test to determine whether the mean absolute percentage error for the mcmcDivE estimate is significantly smaller than those for the other diversity estimation methods in each set of 100 simulations. We additionally computed the coefficient of determination ($R^2$) between estimated and true diversity to assess how closely the estimated trends matched the true diversity trajectories. We compared the performance of the mcmcDivE estimates with a curve of raw sampled diversity (i.e. number of sampled species per 1 Myr time-bin), a range-through diversity trajectory based on first and last appearances of sampled species, and sampling-corrected trajectories estimated using coverage-based rarefaction (*estimateD* function in the iNEXT R package[70]) and the squares extrapolator[88].

From our simulated results, we find that mcmcDivE provides accurate results under most settings and significantly better estimates (significantly smaller mean absolute percentage error; $p < 0.0001$ for all six sets of simulations) of the diversity-through time compared with raw diversity curves, range-through diversity trajectories or sampling-corrected estimates from coverage-based rarefaction, or extrapolation by squares (Fig. 9). The mean absolute percentage error averaged

0.13 (95% CI: 0.04–0.29) in simulations without across lineage rate heterogeneity (Fig. 9E), with a high correlation with the true diversity trajectory: $R^2 = 0.93$ (95% CI: 0.72–0.99). The diversity estimates remained accurate even in the presence of time intervals with zero preservation (Fig. 9B).

Simulations with rate heterogeneity across lineages (Fig. 9D, F) yielded higher mean absolute percentage errors (0.43, 95% CI: 0.24–0.55) while maintaining a strong correlation with the true diversity trajectory $R^2 = 0.95$ (95% CI: 0.85–0.99). This indicates that, while the absolute estimates of diversity are on average less accurate in the presence of strong rate heterogeneity across lineages (in addition to strong rate variation through time), the relative changes in diversity-through time are still accurately estimated. The increased relative error in these simulations is mostly linked with an underestimation of diversity throughout, which has been observed in other probabilistic methods to infer diversity in the presence of rate heterogeneity across lineages (Close et al.[14]). This, however, does not hamper the robust estimation of relative diversity trends using our method (Fig. 9D, F).

After validating the accuracy of the model, regional analyses of Triassic marine diversity were run for 1000,000 MCMC iterations at 1-myr intervals. We summarised the diversity estimates by calculating the median of the posterior samples and the 95% credible intervals.

**Turnover estimation.** Counts of unique taxa within a sample (geographic area or time bin) are a measure of diversity while the degree of taxonomic differentiation between two samples constitutes a measure of turnover. Taxonomic turnover through time, measured by successive comparison of the taxon pools in adjacent time bins, avoids the pitfall of cryptic turnover hidden within diversity or diversification rate curves as high extinction and origination rates will strongly increase taxonomic differentiation through time. We use the modified Forbes index (Forbes*)[89] with relative abundance correction (RAC)[90] as this combination of methods robustly accounts for both incomplete sampling in each sample and differing abundance distributions between a pair of samples, both of which can bias the apparent degree of similarity[90]. RAC is a potentially computationally expensive procedure as it multiplies the number of null trials by the number of rounds of sampling standardisation applied per trial, and because comparison of multiple samples to return a distance matrix becomes exponentially more expensive with each added sample. To address this issue, we implement the RAC-adjustted Forbes* metric (converted to dissimilarity as $1 -$ Forbes*) using an efficient, parallelised C++ function with an Rcpp wrapper in R. We anticipate that our implementation, which performs orders of magnitude faster than the original version, will ease uptake of this method by other palaeobiologists. Occurrences in each region were first binned at stage level, then with twofold subdivision of the Anisian, Ladinian and Carnian and threefold subdivision of the Norian, using the occurrence midpoint ages. RAC-Forbes* dissimilarity was then calculated for each region between successive pairs of time bins with 100 null trials, and 100 sampling standardisation trials at a sampling quorum of 0.5 for each null trial and empirical estimate.

**Reporting summary.** Further information on research design is available in the Nature Research Reporting Summary linked to this article.

## Data availability

All raw and processed data generated in this study are available in the electronic supplement for this paper at https://doi.org/10.5281/zenodo.6477659. Source data are provided with this paper.

## Code availability

PyRate and mcmcDivE are freely available on Github (https://github.com/dsilvestro/PyRate). All scripts used to conduct our analyses are available in the electronic supplement for this paper at https://doi.org/10.5281/zenodo.6477659.

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

## Acknowledgements
Funded in part by NERC GW4+ DTP studentship S100065-138/123 awarded to J.F.S., NERC BETR grant NE/P013724/1 and European Research Council (ERC) Advanced Grant 788203 to M.J.B, and Swiss National Science Foundation grant PCEFP3_187012 and Swedish Research Council grant VR: 2019-04739 awarded to D.S.

## Author contributions
J.F.S. compiled the stratigraphically revised occurrence dataset and designed the custom R functions for spatial standardisation and RAC-Forbes* with C++ implementation. D.S. designed the probabilistic model mcmcDivE for estimating diversity and validated its efficacy using simulations, with additional validation by J.F.S. against shareholder quorum subsampling and squares. J.F.S. conducted all other analyses. J.F.S., D.S and M.J.B. wrote and commented on the paper.

## Competing interests
The authors declare no competing interests.
