## [Peer Review File · Nature Communications]

Global diversity dynamics in the fossil record are regionally heterogeneousReviewers' Comments:

Reviewer #1:

Remarks to the Author:

This paper describes the application of novel Bayesian methods to identify and characterize the evolutionary and ecological impacts of what the Authors call the Carnian Biotic Crisis. Although the results are interesting and potentially important, in my opinion, the paper is not appropriate for publication in Nature. The reason is that the rather complex statistical methods are not what I would describe as well-established - indeed, some are developed as part of the present manuscript. In my opinion, the Authors should first publish the methods developed here in the peer-reviewed specialist literature before submitting a manuscript in which they are applied to data from the Carnian Pluvial Episode. I would also add that it is not enough to show that the methods work well in applications to data simulated under the model on which the methods are based. It is also important to understand how robust the results are to deviations from this model (as many of the model assumptions appear to have been made for mathematical convenience). I do not want to go overboard here: the methods that at least some of the Authors have developed have the potential to be quite powerful in inference based on incomplete fossil data. But there is a need to move a bit more carefully with their actual applications.

Reviewer #2:

Remarks to the Author:

The Carnian Period of Late Triassic represents a time of major climatic changes which attract increasing research attentions. Sutherland et al. investigate the biological and ecological impacts associated with the mid-Carnian climate shift (a.k.a., the Carnian Pluvial Episode) and is a very important contribution to our understanding on the Late Triassic ecosystem evolution. This manuscript could become a good paper after some revisions here and there. I have some minor concerns and suggestions for the authors to consider, listed below, but none of them should tarnish the significance of this paper.

I also notice that there are no line numbers in this manuscript—this makes the comments very difficult. Nevertheless, I hope the authors find my comments helpful to improve the paper further.

Concerns and suggestions:

1. The notion "Carnian Biotic Crisis"

If I am not entirely wrong, the authors use the term for the faunal turnovers in the mid-Carnian. If this is the case, wouldn't "mid-Carnian biotic crisis" be a better and more accurate term? I personally prefer to use the "Carnian Crisis" to bracket all dramatic changes in the Carnian and "CHE" or CPE specifically for the changes in the mid-Carnian. Carnian was a rather long interval of ~10 Ma, almost twice as long as the entire Early Triassic and many aspects are not thoroughly studied. For example, the earliest and latest Carnian are actually poorly understood. The term e.g., "mid-Carnian biotic crisis" will leave some space for future studies to explore and possibly ensure the term is well-accepted and long-lasting.

2. Climate stress in the Tuvanian

Later Tuvanian was probably an interval of high environmental stress which was not well-studied before. This is partially shown in Sun et al. 2016 EPSL that Tuvanian 2 was warmer than the rest of Carnian. As shown in the Fig. 1 of this paper, marine relative diversity is actually very low in the late Carnian. Although this is not the focus of this paper, I wonder if the authors should extend a bit to this aspect.

3. Insightfulness of the discussion

I have to say that I am slightly disappointed by the discussion, particularly on pages 6 and 7. This is probably a very biased opinion because I know the Carnian changes very well. If possible at all, I would encourage the authors to improve the discussion further and bring more new insights into this exciting paper. I am sure that the last author would manage this in no time.

Specific comments:

Page 2, 2nd line, the term "biotic interchange" is not clear to me. Is it faunal turnover, geographic migration or ? Please revise or be more specific.

Page 2 at citation 4, timing of radiations of calcareous nannoplankton is very controversial because they were rather rare in the Late Triassic. I personally think the radiation was in the later Triassic and Jurassic. Please consider revision to avoid potential criticism.

Delete "form the basis of modern marine ecosystems"—the statement was not correct in a strict sense. Cyanobacteria and Diatoms are also important components of modern ocean. One of them occurred long before CPE while the other occurred long after the CPE in the Jurassic.

Page 2 After citation 7, "extreme volcanism"—a usual expression. Consider rephrase.

The following paragraph is rather wordy, better to be concise.

Page 2-3 transition, wordy, consider revise.

Page 6, citation 18—this is too speculative and basically not correct. Reef building release CO₂. I would recommend to delete this sentence since it is not really tightly connected to the CPE and beyond the scope of this manuscript.

Page 7, The Pangaeon climate part is a bit lengthy and not organically bonded with the data of this paper. Please consider to improve.

Fig. 1 The orientation of words in X-axis is very strange, upside down. My neck does not like this layout. Figure caption says carbon isotope curve is from ref 73 but ref 73 is "The igraph software package for complex network research". I would guess it might be 74, but this paper (ref.74) is very problematic, with mismatches in biostratigraphy and many one data point excursions, amongst numerous other problems. I would recommend to be very cautious on these details—they are not key components of this study but could tarnish the overall quality of this paper.

Fig. 2 Names of stages and substages (if possible) should be incorporated into the X-axis. This should also apply to Fig. 3, 4, 6.

Fig. 3 As fig. 2

Fig. 4 If possible, combine it with Fig. 3

Fig. 6 and 7—they are introduced in the results but not mentioned in the discussion. I think they can be better explained and/or better bonded with the discussion.

Yadong Sun

University of Erlangen-Nuremberg

Response to Reviewers

Reviewer comments are in black. Our responses are in red

Reviewer #1 (Remarks to the Author):

This paper describes the application of novel Bayesian methods to identify and characterize the evolutionary and ecological impacts of what the Authors call the Carnian Biotic Crisis. Although the results are interesting and potentially important, in my opinion, the paper is not appropriate for publication in *Nature*. The reason is that the rather complex statistical methods are not what I would describe as well-established - indeed, some are developed as part of the present manuscript. In my opinion, the Authors should first publish the methods developed here in the peer-reviewed specialist literature before submitting a manuscript in which they are applied to data from the Carnian Pluvial Episode. I would also add that it is not enough to show that the methods work well in applications to data simulated under the model on which the methods are based. It is also important to understand how robust the results are to deviations from this model (as many of the model assumptions appear to have been made for mathematical convenience). I do not want to go overboard here: the methods that at least some of the Authors have developed have the potential to be quite powerful in inference based on incomplete fossil data. But there is a need to move a bit more carefully with their actual applications.

We would make a strong case against this opinion. We believe it is appropriate to present a new method (which is not as radically new as the Reviewer implies because it is part of the well-established, and widely used, PyRate package) as part of a case study paper, so the reader can evaluate the method while seeing it in action in a real case. In terms of suitability for *Nature Communications*, we argue this is in line with many recent papers there (including previous additions to the PyRate package), where the paper makes multiple, strong contributions and so attracts wide attention.

The estimation of preservation, origination, and extinction rates presented in our study are based on Bayesian algorithms thoroughly described and tested in several previous articles, including Silvestro et al. 2014 *Systematic Biology*, 2018 *Nature Communications*, and 2019 *Paleobiology*. We therefore consider these methods as well established and do not further validate them with simulations in the present study. We do present a new method to infer corrected diversity trajectories based on sampled diversity and preservation rates inferred by PyRate and we have revised it in two ways following the Reviewer's suggestions.

1. First, we changed the implementation of the model, which was previously based on a rather complex two-step approach including Monte Carlo simulations and Bayesian optimization. This is now simplified to a fully probabilistic approach that takes a single step using a hierarchical Bayesian algorithm to infer diversity through time. While the method is new, it is based on well-established algorithms (Markov Chain Monte Carlo sampling) and likelihood models (binomial sampling probability and Brownian processes; Starrfelt & Liow, 2016 *Phil Trans Roy Soc*; Brocklehurst, 2015 *Palaeo Electronica*). We have now re-written the description of the model in the Methods section of our manuscript and significantly simplified the code implementing it. We believe that the revised version of the manuscript provides a clearer and more transparent explanation of the new method presented here.
2. We added a new set of 100 simulations to account for cases of zero-preservation in some time intervals. In these new simulations, the entire fossil record is removed within two time windows of several millions of years to assess to what extent these settings may deteriorate the accuracy of our diversity estimator. We found that, while gaps in preservation increase the relative error of diversity estimates, this is still low, being limited to 8–15%. The results are presented in the new Figures S4–5

In our revised manuscript, we validate our new method through the analysis of 600 synthetic datasets and under a wide range of settings including different diversification histories, and strongly varying preservation rates through time and across lineages. We think the scenarios considered in our simulations provide a comprehensive evaluation of the strengths and limitations of our method. Our primary results also remain robust despite the change in model implementation, further highlighting both their own validity and that of the model.

Reviewer #2 (Remarks to the Author):

The Carnian Period of Late Triassic represents a time of major climatic changes which attract increasing research attentions. Sutherland et al. investigate the biological and ecological impacts associated with the mid-Carnian climate shift (a.k.a., the Carnian Pluvial Episode) and is a very important contribution to our understanding on the Late Triassic ecosystem evolution.

Thank you for endorsing the importance of the case study.

This manuscript could become a good paper after some revisions here and there. I have some minor concerns and suggestions for the authors to consider, listed below, but none of them should tarnish the significance of this paper. I also notice that there are no line numbers in this manuscript—this makes the comments very difficult. Nevertheless, I hope the authors find my comments helpful to improve the paper further.

We appreciate the further positive comment and have addressed all the detailed corrections. Apologies for no line numbers – these have also been added to the manuscript and are referred here also.

Concerns and suggestions:

1. The notion “Carnian Biotic Crisis”

If I am not entirely wrong, the authors use the term for the faunal turnovers in the mid-Carnian. If this is the case, wouldn't “mid-Carnian biotic crisis” be a better and more accurate term? I personally prefer to use the “Carnian Crisis” to bracket all dramatic changes in the Carnian and “CHE” or CPE specifically for the changes in the mid-Carnian. Carnian was a rather long interval of ~10 Ma, almost twice as long as the entire Early Triassic and many aspects are not thoroughly studied. For example, the earliest and latest Carnian are actually poorly understood. The term e.g., “mid-Carnian biotic crisis” will leave some space for future studies to explore and possibly ensure the term is well-accepted and long-lasting.

Agreed. In earlier papers, people pointed to an end-Carnian crisis, and we must be clear we refer to the mid-Carnian, so we adopt this term, as suggested.

2. Climate stress in the Tuvanian

Later Tuvanian was probably an interval of high environmental stress which was not well-studied before. This is partially shown in Sun et al. 2016 EPSL that Tuvanian 2 was warmer than the rest of Carnian. As shown in the Fig. 1 of this paper, marine relative diversity is actually very low in the late Carnian. Although this is not the focus of this paper, I wonder if the authors should extend a bit to this aspect.

We add consideration of this point in the Discussion. This is probably highly relevant to evidence for later extinction and recovery on land.

3. Insightfulness of the discussion

I have to say that I am slightly disappointed by the discussion, particularly on pages 6 and 7. This is probably a very biased opinion because I know the Carnian changes very well. If possible at all, I would encourage the authors to improve the discussion further and bring

more new insights into this exciting paper. I am sure that the last author would manage this in no time.

We have revised the Discussion, especially by adding consideration of the climate changes on terrestrial floras and faunas, and by adding some more on the Tuvlian climate point.

Specific comments:

Page 2, 2nd line (line 37-38), the term “biotic interchange” is not clear to me. Is it faunal turnover, geographic migration or ? Please revise or be more specific.

Wording has been revised to ‘faunal turnover’ for accuracy

Page 2 at citation 4 (line 46), timing of radiations of calcareous nannoplankton is very controversial because they were rather rare in the Late Triassic. I personally think the radiation was in the later Triassic and Jurassic. Please consider revision to avoid potential criticism.

This has been revised to the ‘origin of calcareous nannoplankton’ to better reflect the timing of events in their fossil record – i.e. their origination in connection with the Carnian Pluvial Episode which is supported by current evidence, but not their taxonomic radiation which, as Reviewer 2 points out, occurred later.

Delete “form the basis of modern marine ecosystems”— the statement was not correct in a strict sense. Cyanobacteria and Diatoms are also important components of modern ocean. One of them occurred long before CPE while the other occurred long after the CPE in the Jurassic.

This has been changed to ‘key components of modern marine ecosystems’, (line 46) which is true, but does not convey the exclusivity that the previous wording possessed.

Page 2 After citation 7 (line 51), “extreme volcanism”—a usual expression. Consider rephrase. The following paragraph is rather wordy, better to be concise.

This has been changed to elevated volcanism, both a different expression and a more conservative option given continued uncertainty regarding the magnitude of volcanism associated with the Carnian Pluvial Episode. The wording of the following paragraph has also been revised for clarity.

Page 2-3 transition, wordy, consider revise.

This has been streamlined.

Page 6, citation 18 (line 177-178) —this is too speculative and basically not correct. Reef building release CO₂. I would recommend to delete this sentence since it is not really tightly connected to the CPE and beyond the scope of this manuscript.

Our original wording of 'linked to' was ambiguous as the cited paper was saying that the acquisition of symbionts may have been 'a response' to a Carnian-Norian drop in CO₂, not 'the cause of' as the reviewer may have interpreted, given our original wording. We have revised this point to clarify that corals may have acquired symbionts in the Carnian but have removed any reference to climatic effects, thus keeping the focus purely on biotic turnover in response to the Carnian Pluvial Episode.

Page 7 (line 214), The Pangaeian climate part is a bit lengthy and not organically bonded with the data of this paper. Please consider to improve.

We have addressed this issue by referring to our results (the palaeogeographic map displaying extinction intensities) at the beginning of the paragraph, then linking these results to Pangaeian climates. This section was also streamlined to remove extraneous detail that was not organic within the context of the paragraph.

Fig. 1 The orientation of words in X-axis is very strange, upside down. My neck does not like this layout. Figure caption says carbon isotope curve is from ref 73 but ref 73 is “The igrph

software package for complex network research". I would guess it might be 74, but this paper (ref.74) is very problematic, with mismatches in biostratigraphy and many one data point excursions, amongst numerous other problems. I would recommend to be very cautious on these details—they are not key components of this study but could tarnish the overall quality of this paper.

We realised that the position in which we originally attached the figure was misleading. We had rotated the figure to maximise use of the page and give the reviewers a larger image to view. The figure is meant to be read in landscape format where the orientations of the axes are conventional – we have updated the manuscript accordingly.

Regarding the isotopic data, we have retained the original curve as it shows the extent of the Mid-Carnian negative isotope excursion within a broader temporal context, demonstrating the distinct geochemical signal of the event. To address concerns over the quality of the data itself, we have added an additional carbon isotope curve taken from a shorter duration, but extremely well resolved section. This high-quality curve (ref. 33) records the same trends of chaotic carbon fluctuation across the Carnian Pluvial Episode, bolstering the more coarsely resolved, yet temporally broader curve taken from reference 74. The figure caption references have been updated accordingly.

Fig. 2 Names of stages and substages (if possible) should be incorporated into the X-axis. This should also apply to Fig. 3, 4, 6.

Fig. 3 As fig. 2

It was not possible to incorporate stage and substage names into the plots without significantly reducing their clarity. As an alternative, the legend of the figure where the full scheme of coloured bars denoting stages/substages is introduced has been updated to refer the reader to Table 1 for guidance, where the divisions are listed in the same order.

Fig. 4 If possible, combine it with Fig. 3

These figures have been combined. As above, however, this did not leave enough space to add stage and substage names into the plots without significantly reducing their clarity

Fig. 6 and 7—they are introduced in the results but not mentioned in the discussion. I think they can be better explained and/or better bonded with the discussion.

We have expanded on these results in the Discussion to further show how they demonstrate biotic turnover in response to the Carnian Pluvial Episode in terms of ecological restructuring. We anticipate that this will also help address Reviewer 2's comments regarding the depth of the Discussion.

Reviewers' Comments:

Reviewer #1:

Remarks to the Author:

I have not changed my opinion - and it is only an opinion - about the suitability of this manuscript for publication in Nature. My concern is that it is too heavy on statistics. In my review of the previous version, I referred to complex methods that were not well-established. I should have been clearer that the the latter was intended to apply to the new method described in the manuscript rather than (or at least less) to the methods previously published by the Authors. The fact that the Authors were able to simplify (to a degree) the former in the revised manuscript reinforces (to me) the need for further elucidation and testing. As I noted in my previous review, both the published method and its extension seem strongly parametric, raising a question about robustness. I do want to repeat my previous comment that I mean no denigration. However, it is my (possibly false) impression that Nature is not the place for a paper the substantive results of which depend on a complex statistical method at this stage of development.

Reviewer #2:

Remarks to the Author:

Dear Editor and authors,

Climate changes in the Carnian, Late Triassic, is no doubt enigmatic. They were accompanied with profound faunal and floral changes, which were not properly addressed previously. My opinion on this manuscript remains unchanged—Sutherland et al. delivered a nicely-addressed paper, which focuses on ecosystem changes during the Mid-Carnian crisis using quantitative palaeobiological assessment. This revision shows significant improvements upon the last version. I am pleased to see many changes, bringing in new insights to our understandings on the Carnian Crisis in general. Concerning the criticism on the method by Reviewer #1 in the last round of review, I cannot give an in-depth assessment since it is beyond my field of expertise. As a general audience of this field, the methodology of this paper is convincing.

In summary, a minor revision is recommended before the publication of this manuscript. I have some comments here and there for the authors to consider. They are only my opinions and the authors do not have to agree with me.

General comments:

1. Ma v.s. Myr: In a very strict sense, Ma should be used as time point (e.g., 253 Ma) while Myr should be used for time interval (e.g., ~1 Myr duration). This has not been done consistently throughout the manuscript. If the authors agree with me, kindly check this throughout the manuscript (e.g., lines 42, 43 v.s. e.g., line 250).
2. Since this manuscript will likely become an important paper in this field. I encourage the authors to include a few words about future research directions and/or the limitation of the approach applied. One or two sentences will probably be enough. This is partly done by the authors in lines 186-193. Words as such will give audience opportunities to re-think the entire Carnian biotic changes and probably inspire future works.

Specific comments:

Lines 78-81, The method is indeed very superior. Age assignments of PBDB entries, as a matter of fact, are very worrying. Just out of curiosity, would this method somehow tolerate e.g., 15-20% incorrect data entry?

Line 117, not very clear here. Last time bin in the Late Triassic, the entire Triassic or?

Line 151, please pardon my ignorance, I am not familiar with the term "network communities". Maybe give a few examples to improve readability.

Line 227, What does "its" refer to here?

Line 264, diatoms were probably not amongst these—better delete. References cited here do not have any information for diatoms at this time.

Line 296, two words are in red.

Comments on figures:

Fig. 1 Because the upper part of this diagram is in different time scale with the lower part of this diagram, names of stages for the latter part should also be included to improve clarity.

Fig. 2, 3, and 5 I would still recommend to include stage (substage) names on X axis. It is not difficult to do in R but will significantly improve readability. If this is somehow not possible, abbreviations may be possible to included manually.

Fig. 6 Not sure what is the problem, the PDF file I have should an overlap of Fig. 6 and Table 1 (page 27).

Yadong Sun

Reviewer #3:

Remarks to the Author:

The great thing about Bayesian modelling is that the story is easy to tell. You can start by explaining your notation, then write out the posterior distribution you want to get, and then show the likelihoods and the priors. In most models of reasonable simplicity this can be done in few a few paragraphs and the whole model is clear and easy to understand. The authors really haven't done a good job here of explaining this model and, as someone who works on a daily basis with Bayesian models of all types, this is not easy to follow. The methods section needs to be written much more clearly before it can even be reviewed, let alone accepted.

As somebody most interested in the methods, I jumped straight to that section to try and understand the basics of the model. We are first told that t is time and q_t is 'the expected number of fossil occurrences per lineage per time unit (typically 1 myr) averaged across all lineages present at time t '. It would be nice to have an equation for this, and also find out whether this is a parameter to be estimated or data included in the model (we find out later). My guess is that something (let's call it Y_t) is Poisson distributed with rate q_t . We are then told that q_t is Gamma distributed with rate and shape parameters α . It's not clear why these would be set equal and seems like a funny modelling choice. In the next paragraph we are told the details of the MCMC algorithm, despite the fact that we haven't even got an explanation of the notation, let alone the model. There is an obscure reference to an 8-category gamma distribution (an 8-dimensional multivariate gamma distribution, a gamma distribution with 8 different values of rate/shape/both, a hierarchical formulation?). Alas we are never given enough detail to form an opinion.

In the section 'Probabilistic diversity estimation' we are at last presented with some clear equations. We are given n_t as the number of sampled taxa which is binomial with probability t and total number of species $D_t = n_t + x_t$ where x_t is the missed species. This looks just like an N-mixture model as is often used in statistical ecology (though I can't see that referenced). Finally, we are given a posterior:

$$p(D \mid s^2, n, q, \alpha) \propto p(n \mid D, q, \alpha) \times p(D \mid s^2) p(s^2)$$

We are given the prior for D (with some very mangled notation and missing close-brackets) but we have to guess at the likelihood (something to do with the binomial distribution specified above?) and the prior for σ (which I wrote as s above for brevity).

More deeply, it seems very unsatisfactory that the 'data' here, i.e. n , q , and α , aren't really data at all but outputs from other models (PyRate). It's not clear why this couldn't have been fitted in one step and just have one single overall model which explains the lot.

As a final minor point, it seems a tad strange for such a simple model (note I am not criticising the

model for being simple - I like simple models) not to have been fitted in one of the many standard tools (Stan, JAGS, Nimble, many of which have both Python and R versions) to save them the trouble of coding this all up by hand. My guess is that a Hamiltonian Monte Carlo version of this model you would need only a few thousand iterations and nothing like the million plus samples that the authors use here.

□□□□□□□□□□ □□□□□□□□□□ □□□

Reviewer #4:
Remarks to the Author:
General Comments:

The Carnian-Pluvial Event has attracted a growing amount of attention within recent years, and as such a quantitative study assessing the potential impacts of the CPE on marine and terrestrial ecosystems is a more than welcome addition to the literature. The authors perform an extensive list of tests with the aim of resolving the effects of the CPE on diversity, extinction and origination rates and ecological networks. They conclude that the Mid Carnian Biotic Crisis resulted in enhanced turnover and large changes in community ecology within both marine and terrestrial settings. Whilst this topic is well suited for the remit of this journal, there are a number of serious concerns I have regarding both the interpretations of the data that prevent me from recommending it for publication at this time.

The authors carry out a great number of tests to quantify the effects of the Carnian Pluvial Event. However, many of these tests, in my opinion, do not show the the Carnian Pluvial Event as being a major mass extinction and I do not feel that the specific interpretations of the impacts of the CPE are backed up by the data presented. In almost all of the figures presented for this paper, equally large or larger changes are observed in either the preceding or following time bins. For example, an equally large drop in marine diversity is found within the Early Carnian (~243 mya) in probabilistic and traditional SQS methods; larger changes in all network analysis approaches are observed between 250-240 mya for both marine and terrestrial organisms; and extinction and origination rates show extinction drastically outpacing origination within the Late Norian and Rhaetian, to a degree that is not seen across the Carnian. In fact, both probabilistic and traditional diversity metrics report an increase in terrestrial diversity across the CPE. If I was shown the various data presented in this study without prior knowledge of the CPE, I would be more interested in examining the events before and after this event, as they show greater volatility and change. Whilst the authors acknowledge that there appears to be a lack of impact on terrestrial ecosystems and comment that 'the taxonomic and ecological effects of extinction are just as important as the magnitude', there is no methodological attempt to dig into why there might be a disparity here. Additionally, the following argument that terrestrial networks were significantly impacted across the CPE which is highlighted within the abstract ignores the fact that the drop in network transitivity is caused by a fall from an anomalous spike in the Early Carnian, before the CPE, and is returning to pre-spike levels. To me, this pattern is more interesting to following than any changes across the Early-Late Carnian boundary. As such, it appears to me that the authors are already convinced of the CPE as an event that caused major impacts within ecosystems, and are selectively interpreting data to fit this narrative. This also makes it very hard to comment on the discussion, which builds off the authors' interpretations. This paper would be far stronger if the data was presented neutrally and reasons for disconnect between this and prior studies were dug into more deeply.

Another serious issue is related to the lack of investigation around other potential causes of the patterns observed here. Biases arising from spatial variation in sampling have increasingly been seen as important for understanding diversity and ecology in deep time. Changes in the amount and location of sampled outcrop between stages has the potential to be viewed as extinction events or major changes in ecological structure rather than a bias arising from geological and anthropogenic means. Consequently, there have been a multitude of papers within the last few years that discuss the

spatial biases in the fossil record and how to counteract them (e.g. Benson et al. 2016, Close et al. 2017, Dean et al. 2019, Close et al. 2020a,b). These issues are not addressed or mentioned within this manuscript, which seems like a major oversight for a study that uses diversity curves. Whilst I understand that the probabilistic methods here intrinsically incorporate rates of sampling, these can still be affected by the underlying change in location of the sampling pool. Currently, it is impossible to evaluate whether patterns like the supposed gap between northern and southern communities in the Late Carnian are a result of genuine ecological signals, or a lack of sampling opportunity between the locations. The same is seen for extinction intensities (Figure 4); it is hard to know whether the Late Carnian genuinely shows an increase in extinction intensity, or just an increase in sampled area compared to the other stages (or vice versa, whether the other stages would show equally as intense extinction intensities if it was possible to sample more cells). This section could be drastically improved by implementing a geographic subsampling approach that randomly selected a certain number of geographic cells, and I would be a lot more comfortable discussing the results if these kind of measures had taken place. There are also temporal biases at play. Although I am not the most familiar with this period of geological history, I know that obtaining accurate chronostratigraphic ages for formations and fossil occurrences at this time is problematic, and that there is a reasonable amount of stratigraphic uncertainty. In fact, the authors themselves mention this within the paper (P6, Line 191). Whilst the authors compiled new ages for formations in the dataset, this is not really discussed in the context of the results, and it is unclear what knowledge this gave the authors of the stratigraphic uncertainties associated with this time period. Discussion of this uncertainty and the limitations of the data would strongly improve the manuscript and give greater confidence in interpretations.

As a final point, whilst I cannot comment on the validity of the new probabilistic diversity method in terms of a technical standpoint, I am almost in agreement with the previous reviewer who commented that it would be better the method be presented in a separate study. Whilst I do not agree that it is necessary to have a completely separate study to implement a new technique, I do find it hard to appropriately assess the severity of the drops in relative probabilistic diversity when there is no appropriate comparison with other major extinction events. I would be more convinced about the severity of the Mid-Carnian Biotic Crisis if I could compare diversity curves with those run over the P-T extinction, or the T-J extinction. I understand this would potentially require a substantial amount of additional work, but it would provide good evidence of the impact, or lack of impact of the CPE.

I don't want these general comments to be entirely negative – the authors have done a great job tying together a number of complex, intricate analyses carried out on an enormous dataset, and the manuscript itself is clear and well written. The extent of the supplementary data also shows the lengths that the authors have gone to to carry out clear and well documented science. Additionally, whilst my knowledge of Bayesian approaches is limited, the newly documented method for diversity estimation looks interesting, and is sure to be something a great number of palaeontologists and ecologists will be interested in. As such I feel this work fits well within the remit of Nature Communications. However, I feel that the work cannot be published without a more balanced discussion of the data presented and its limitations, and without additional measures taken to account for spatial biases in the fossil record. My overall recommendations for the authors would be to simplify the study, and focus on a few results (for instance the probabilistic diversity curves with the addition of spatial sampling) in greater depth.

I have some other comments which I hope the authors find helpful for improving the manuscript.

Line by line comments:

INTRODUCTION

P2. Line 36. "It is now recognised..." I would not count a single paper as consensus. As such, I would recommend changing this to "A recent study has suggested..."

P2. Line 43. "Widespread biotic turnovers..." This sentence needs a citation.

P2. Line 48. "The CPE marks a revolution..." This is a substantial claim that requires substantial evidence. I would request that either explicit reference to several, well regarded studies are added or that this sentence is toned down in the statements that it is making.

P2. Lines 53-54. This succinctly sums up an issue with the introduction – the first paragraph repeatedly claims that the CPE fundamentally revolutionised marine and terrestrial ecosystems, yet in the next paragraph it is shown that there have been minimal quantitative assessments of its biotic impacts. Which is it? I recommend toning down the statements in the first paragraph to more accurately reflect the state of prior and current knowledge on the CPE.

P2. Line 59. "To which the Mid-Carnian Biotic Crisis is now compared". Once again, I think this is an over sell of the situation that requires more than one reference.

P2, Line 61. "... are not overshadowed by the Big Five, which otherwise dominate Phanerozoic diversity curves". This is an odd way of phrasing the situation - it almost sets up a 'cover-up' of the MCBC as an event due to the big five extinctions . I would suggest finding a fairer way of comparing these events (e.g. "without assessing the Mid Carnian Biotic Crisis within an appropriate temporal context, its impacts cannot be adequately compared to other major mass extinctions throughout geological history").

P3. Line 73. "...time binned data". If I'm not mistaken, even the probabilistic methods you employ use time bins of 1 ma. I would recommend changing this to "coarse time binned data" instead.

P3. Line 75. "We are developing..." This reads slightly odd. Developing as in not yet finished? Or developing as in used in this study? Or developing as in methods continue to develop? Please clarify this statement.

RESULTS

P3. Lines 90-92. It may be true that there is a decline, but there is an equally large decline in marine diversity at approximately 242 Mya during the Julian. As an additional point, it is difficult to assess the severity of this decline in comparison to other major extinctions when this is a newly applied method. Is there any possibility of applying this approach to another major mass extinction for comparative purposes?

P3. Lines 93-95. Once again, there is an even larger drop in these curves from the Ladinian to the early Carnian, yet that is not being called a mass extinction. These data needs to be presented fairly.

P3. Lines 100-109 Again, whilst origination rates dip marginally below speciation rates in the Late Carnian, they do so to a fairly similar amount from the Anisian to the Ladinian, and even more so towards the latter stages. To me, this signifies that the Carnian is really not that unique in comparison.

P4. Line 110. It appears to me that there is a data issue here, as these figures show extinction and origination rates which are suspiciously flat for the majority of the Triassic. It strikes me as very odd that these rates are completely flat for the majority of the time, and only show changes at specific points that appear to match fairly closely with either the centre or edges of time bins. As such, I wonder if this is an edge effect potentially related to time binning issues arising from the majority of

organisms being associated with the midpoints of a few formations and/or bins. This is also seen succinctly in the graphs on the right hand side, which all show individual major peaks in rate change. However, even ignoring this issue, these rates show limited changes across the CPE/Mid to Late Carnian boundary in comparison to much larger changes prior and post the CPE.

P4. Line 117. Something that immediately jumps out to me here is the use of the mean as a cut off point for values highlighted in bold – whilst you mention elsewhere the extremely high values in the Induan are ignored, there are equally high values for some other states that could weight the mean. I would recommend using the median as a measure of central tendency instead.

P4. Lines 132-137. I have an issue with the use of the blue colouration on these graphs and the description of this figure within the figure caption. You state that 'colourless cells have no sampled data, while blue cells have no tallied extinctions'. I would be stunned if each of these time bins had occurrences in exactly the same grid cells, and from figure 6 it seems unlikely that is the case. Obviously I cannot be sure as these points represent formations rather than occurrences, but on a check on the PaleoBioDB navigator, there are clearly cells area of the globe which contain no collections within the Ladinian but do within the Carnian. As such it seems more likely that the blue cells represent the total sampled area across all time intervals. This is problematic as it makes it appear as if sampled area is equal through time, when it most definitely is not. This has ramifications on interpreting the spatial sampling of your data, and whether patterns might derive from geographic shifts in sampling rather than genuine biological signals. This issue must be corrected and the figure caption updated so that readers are not misled as to what the figure actually shows.

As an additional point, I would be interested to know how many of the organisms going extinct are temporal 'singleton' taxa, i.e. originate and become extinct within the same stage. Obviously some of these may be genuine originations and extinctions, but it is also likely that some may be caused by a lack of sampling in the following stage.

P5. Line 139. I am unsure what is being shown within Figure 5A. To my knowledge, the Forbes index compares species-site matrices to estimate the similarity or difference between ecological communities. In the figure caption you say Figure 5A shows "Change in Forbes2 ecological similarity". In the methods section, you show say Forbes2 is used to calculate "Changes in ecological similarity between successive pairs of time bins throughout the Triassic". If this is the case, what are the scores on the figure supposed to show? Are the scores within each bin the comparison between the and the previous bin? To me it appears as if a time series graph is not the best method to present these data. I would recommend a similar approach to Brocklehurst et al. (2017) instead.

However, more broadly I feel there is a potential issue with how Forbes2 is being implemented here. You state within the methods that you binned occurrences within all time bins that they potentially appeared within due to stratigraphic uncertainty. However, you are now using Forbes2 to compare ecological similarity between successive time bins. In which case, are you increasing overall ecological similarity between stages which show a greater degree of stratigraphic uncertainty due to the same occurrences being placed in both bins? This issue needs to be clarified and addressed before these results can be properly analysed.

P5. Lines 139-141. Here you state that Figure 5A shows increasing similarity from the early Carnian through late Norian. I cannot agree with this interpretation - both the marine and terrestrial lines show a basically flat profile, with no obvious trajectory. I would also argue that the alternating pattern described in lines 140-141 can also be extended to the Early and Late Carnian.

P5. Lines 141-143. I would not draw the same conclusions from these data. Once again, more significant changes in all measures appear to occur within the Ladinian – Early Carnian.

P5. Lines 151-159. For Figure 6, I am unsure what the size of communities is showing here – is this

just a measure of alpha diversity? Or is it a measure of community complexity? There needs to be better setup and explanation of null hypotheses here for readers to be able to accurately interpret the data. Also, how are these values drawn on the maps? Formations normally have a wide geographic spread – how were the points picked?

P5. Lines 155-157. I am unsure how these show distinct northern and southern communities, when there are clearly lines connecting between northern and southern regions, more so than in the Early Carnian. Or is this a comment on the fact that there are no communities between the northern and southern regions? In which case, this could also be due to a lack of sample-able rocks within this region. This needs to be more clearly explained.

DISCUSSION

As mentioned above in my general comments, my line-by-line comments on the discussion are limited by the fact that the discussion relies on interpretations of the data that are in my opinion not valid. As such, I cannot adequately assess whether claims made seem legitimate.

P5 Line 167-169. I struggle to believe this interpretation, when as mentioned above there overall seems to be very little change across this boundaries within the majority of the data presented, especially compared to fluctuations in the various curves both before and after the Carnian. For instance, Terrestrial fauna show extinction rates that far outstrip origination rates within the early Norian, much more so than across the Carnian. Why is this not viewed as an even greater, hidden mass extinction?

P6. Line 171. "...in keeping with the dearth..." There should be a citation for this statement.

P6. Line 197-198. This distribution could also be due to a lack of sampled fauna within the tropics, either from a lack of rock outcrop or modern worker effort. Currently it is not possible to distinguish between these hypotheses. As such, I would recommend incorporating this potential argument into the discussion.

P6. Line 199. I personally would not count the network analysis as strong evidence – potential evidence seems more justifiable given the concerns listed above.

P7. Line 227. This is not accurate – the paper in question hypothesised: "This westward migration of its main depocenter could be explained by the Ladinian humid interval (Bernardi et al., 2018) and later Carnian pluvial events (Hochuli and Vigran, 2010) as possible climatic drivers that facilitated resurgence in sediment supply to the basin." This was not tested or examined further. As such, I would recommend correcting this language to reflect this more accurately.

P7. Lines 227-230. This is a great idea, and would make a very neat case study project that could involve a nice bit of fieldwork, sedimentology and analytical work. I would love to see the outcomes of this!

P8. Lines 237-240. The end part of this sentence ("for example from the role in continentality in the persistence of desert conditions") doesn't quite feel like it fits on to the prior sentence – maybe it's just me, but I think this could be clarified.

METHODS

P9. Lines 292-294. Age constraints for formations are an incredibly important part of palaeontology that is often forgotten, so I am glad to see this being carried out. However, it would be good to have some additional information. Were these ages updated to have specific ages for the starts and ends of formations? Or were they updated to fit within the time bins chosen within this study? I had a look in

supplementary information and could not find a record of a list of formations, so I assume it is the latter. If so, it would be interesting to know a breakdown of how many formations can be actually split into the Early or Late Carnian, and how many cross over that interval.

P9. Lines 297-299. I would be interested to know the difference between these datasets – how many occurrences were retained that had poor time constraints? How badly were the data affected? You have already made reference to the fact that tightly controlled time intervals are important. Providing these kind of confidence intervals is crucial for readers to be able to accurately interpret your data.

P9. Lines 299-301. Am I correct in saying that you binned occurrences within each bin they potentially occurred within due to stratigraphic uncertainty? If so I would just say this – it's now fairly standard practise for diversity studies (either that, or randomly placing occurrences within a bin and repeating to gain an average, or only placing within the majority bin).

P10. Line 328. Why quorum of 0.8? This seems unusually high for a deep-time diversity study, and I am surprised you can get accurate curves from this. Did you attempt to see patterns at lower quora?

P11. Lines 354-356. You state here: "Marine and terrestrial analyses were run for 30 million MCMC generations, sampling every 5000 generations, while terrestrial analyses were run for 20 million generations". Does this mean terrestrial analyses were run for 20 million more? Or is this a typo, and were marine run for 30 million and terrestrial for 20 million?

P13. Line 408. Please excuse this if I have misunderstood the method, but I have a slight issue with the high temporal resolution used here. From my understanding, the method calculates the relative diversity at each 1 ma time bin of a total 50 time bins based on the fossil occurrences present and a calculated sampling probability. However, on a check of the full dataset in R, there are only about 15 unique ages associated with the dataset. Isn't there therefore a large discrepancy between the resolution of the analysis carried out in comparison with the occurrence data available? I think my concerns would be lessened if you presented an overview of the temporal limits of formations and occurrences, so that I could see the comparison between the different resolutions.

P14. 445. You say here that you removed poorly sampled formations with fewer than 2 collections (i.e. 1 collection). For ecological analyses, only removing formations with 1 collection seems lenient. For a similar study, Vavrek and Larsson (2010) removed any formations with fewer than 100 specimens. I would therefore recommend your rationale for only removing formations with 1 collection, and not say 5-10.

Page 14. 455-458. These terms are listed, but not explained in terms of what they actually mean (i.e. a value of X means Y). It would help the reader to have clear definitions of what each of these metrics means in terms of ecology.

FIGURES

As a general comment, it would be extremely helpful to have the stage designations, or chosen bins that are discussed in methods on graphs, and not just the grey boxes indicating stages. It is currently difficult to assess which bin is which bin without going back and forth between the figures and the methods. This extends to Figures 1, 2, 3 and 5.

Figure 1.

Figure 1 is beautiful – such a wonderful way to present data, and the colour scheme is great.

Figure 3.

For Figure 3, the y axis on graphs B and F says Speciation Rate, whereas the figure caption says origination rate. This might cause some confusion in readers – I would suggest consistency, or stating

somewhere in the manuscript that these terms are used interchangeably for this work.

Figure 4.

As mentioned in the comments above, I would recommend removing any cells that contain no genera for each time bin so as to reduce potential confusion for readers. This will also more effectively show the global distribution of sampling for each stage. Additionally, I think that the shaded shallow marine area (I presume it is that?) is not a good addition – it's hard to judge what the actual colour of some of the hexagons is due to this. As such I would recommend removing this if possible. I would also recommend reducing the transparency, and making the colours more opaque so that differences between extinction intensities can be more easily recognised.

Figure 6.

The size of communities in these maps is hard to assess due to the lack of a gradational colour scheme. I would strongly recommend changing this to aid readers. Additionally, the grey colour of the minimum spanning trees is almost the same as the colour for shallow marine area. Please change this so that the two things are most easily differentiated.

TABLES

Table 1 and 2

Caption at the top reads L CRN and U CRN (lower and upper). This is distinct from Early and Late – the two have different stratigraphic meanings. Please correct this so that they are reported consistently throughout the manuscript.

After addressing the concerns and suggestions above, I feel that the manuscript will contribute a significant impact to palaeontological studies and will be suitable for publication.

If you have any comments or queries, I will be happy to answer them for you – please just shoot me an email. (christopherdaviddean@gmail.com)

Best wishes,

Christopher Dean.

References:

BENSON, R. B., BUTLER, R. J., ALROY, J., MANNION, P. D., CARRANO, M. T. and LLOYD, G. T. 2016. Near-stasis in the long-term diversification of Mesozoic tetrapods. *PLoS biology*, 14, e1002359.

BROCKLEHURST, N., DAY, M. O., RUBIDGE, B. S. and FRÖBISCH, J. 2017. Olson's extinction and the latitudinal biodiversity gradient of tetrapods in the Permian. *Proceedings of the Royal Society B: Biological Sciences*, 284, 20170231.

CLOSE, R. A., BENSON, R. B. J., ALROY, J., CARRANO, M. T., CLEARY, T. J., DUNNE, E. M., MANNION, P. D., UHEN, M. D. and BUTLER, R. J. 2020a. The apparent exponential radiation of Phanerozoic land vertebrates is an artefact of spatial sampling biases. *Proceedings of the Royal Society B: Biological Sciences*, 287, 20200372.

CLOSE, R. A., BENSON, R. B. J., SAUPE, E. E., CLAPHAM, M. E. and BUTLER, R. J. 2020b. The spatial structure of Phanerozoic marine animal diversity. *Science*, 368, 420–424.

CLOSE, R. A., BENSON, R. B., UPCHURCH, P. and BUTLER, R. J. 2017. Controlling for the species-area effect supports constrained long-term Mesozoic terrestrial vertebrate diversification. *Nature communications*, 8, 15381.

DEAN, C. D., ALLISON, P. A., HAMPSON, G. J. and HILL, J. 2019. Aragonite bias exhibits systematic spatial variation in the Late Cretaceous Western Interior Seaway, North America. *Paleobiology*, 45, 571–597.

VAVREK, M. J. and LARSSON, H. C. 2010. Low beta diversity of Maastrichtian dinosaurs of North America. *Proceedings of the National Academy of Sciences*, 107, 8265–8268.

Response to reviewers

Reviewer comments are given in regular text, editor comments are given in *italics*, and our responses are in red.

Reviewer #1:

I have not changed my opinion - and it is only an opinion - about the suitability of this manuscript for publication in Nature. My concern is that it is too heavy on statistics. In my review of the previous version, I referred to complex methods that were not well-established. I should have been clearer that the latter was intended to apply to the new method described in the manuscript rather than (or at least less) to the methods previously published by the Authors. The fact that the Authors were able to simplify (to a degree) the former in the revised manuscript reinforces (to me) the need for further elucidation and testing. As I noted in my previous review, both the published method and its extension seem strongly parametric, raising a question about robustness. I do want to repeat my previous comment that I mean no denigration. However, it is my (possibly false) impression that Nature is not the place for a paper the substantive results of which depend on a complex statistical method at this stage of development.

As before, and still more so given publications made since 2019, PyRate is already a widely used and well-established method. Alongside validations using simulations in the original publication on the rjMCMC implementation we use (Silvestro et al. 2019) further simulation work has found that the method remains robust over a wide range of cases (Warnock et al. 2020) and we explicitly discuss the suitability of the method for our application in the text (lines 528-553).

We appreciate that the new method mcmcDivE is also statistically complex. However, we note that it has also been well validated using simulations (supplementary information) and that the recovered diversity curves are congruent with the modelled diversification rates produced by PyRate, giving them strong credibility.

Finally, we note that the editors previously considered publication of the method suitable in this context and continue to believe that method and application should be published together:

In general, we do not agree with Reviewer #1's argument that new methods should be published separately.

However, both Reviewers #3 and #4 raise related concerns about the completeness of the description and the validation of the new method, which convinces us that there is a problem with trying to present the method and use it in a case study in this particular instance. Reviewer #4 also raises important concerns about the interpretation of the results and the possibility of alternative explanations, which would require a much more substantial investigation of the case study even once the method has been better established.

We address the completeness of the description and validation of mcmcDivE in the responses below. We have also changed the case study in which the method is given empirical context and have substantially revised the results and interpretations pertinent to the original manuscript regarding diversification rates at the Carnian Pluvial Episode.

Reviewer #2 (Yadong Sun):

Climate changes in the Carnian, Late Triassic, is no doubt enigmatic. They were accompanied with profound faunal and floral changes, which were not properly addressed previously. My opinion on this manuscript remains unchanged—Sutherland et al. delivered a

nicely-addressed paper, which focuses on ecosystem changes during the Mid-Carnian crisis using quantitative palaeobiological assessment. This revision shows significant improvements upon the last version. I am pleased to see many changes, bringing in new insights to our understandings on the Carnian Crisis in general. Concerning the criticism on the method by Reviewer #1 in the last round of review, I cannot give an in-depth assessment since it is beyond my field of expertise. As a general audience of this field, the methodology of this paper is convincing.

We thank Dr Sun for his positive response to the manuscript. While the focus of the piece has been substantially revised, we hope that it remains a worthwhile contribution to the current literature on the Carnian Pluvial Episode, particularly regarding the continued uncertainty regarding the global extent of its biotic impact.

Reviewer #3:

The great thing about Bayesian modelling is that the story is easy to tell. You can start by explaining your notation, then write out the posterior distribution you want to get, and then show the likelihoods and the priors. In most models of reasonable simplicity this can be done in few a few paragraphs and the whole model is clear and easy to understand. The authors really haven't done a good job here of explaining this model and, as someone who works on a daily basis with Bayesian models of all types, this is not easy to follow. The methods section needs to be written much more clearly before it can even be reviewed, let alone accepted.

Despite the change in paper focus, we retain the two Bayesian methods used in the original submission. The first is PyRate, which jointly models extinction, speciation and sampling rates. The second is mcmcDivE, which models corrected diversity using the same input data as PyRate, along with preservation rate outputs from a PyRate analysis. We apologise for the lack of clarity surrounding model descriptions. To make the distinction between the two models clearer, they are detailed in separate sections as in the original submission, and we have double checked the notation and simplified the description of the mcmcDivE model substantially.

As somebody most interested in the methods, I jumped straight to that section to try and understand the basics of the model. We are first told that t is time and q_t is 'the expected number of fossil occurrences per lineage per time unit (typically 1 myr) averaged across all lineages present at time t '. It would be nice to have an equation for this, and also find out whether this is a parameter to be estimated or data included in the model (we find out later). My guess is that something (let's call it Y_t) is Poisson distributed with rate q_t . We are then told that q_t is Gamma distributed with rate and shape parameters α . It's not clear why these would be set equal and seems like a funny modelling choice. In the next paragraph we are told the details of the MCMC algorithm, despite the fact that we haven't even got an explanation of the notation, let alone the model.

These comments relate to the PyRate model, which fully described in its original publications (Silvestro et al. 2014, 2019). These are referenced in the text should the reader want to delve into the model structure, so we have removed all of the notation described above to aid the clarity of the text, and as they are extraneous given the supporting the references.

There is an obscure reference to an 8-category gamma distribution (an 8-dimensional multivariate gamma distribution, a gamma distribution with 8 different values of rate/shape/both, a hierarchical formulation?). Alas we are never given enough detail to form an opinion.

The gamma distribution is a component of the PyRate model and is described in the references. We have clarified the specific formulation of the gamma distribution we utilise, along with justifying references (line 576-581). Further, we have carefully reviewed and updated our description of the model settings and analytical implementation of the PyRate

model as a whole.

In the section 'Probabilistic diversity estimation' we are at last presented with some clear equations. We are given n_t as the number of sampled taxa which is binomial with probability t and total number of species $D_t = n_t + x_t$ where x_t is the missed species. This looks just like an N-mixture model as is often used in statistical ecology (though I can't see that referenced). Finally, we are given a posterior:

$$p(D \mid s^2 \mid n, q, \alpha) \propto p(n \mid D, q, \alpha) \times p(D \mid s^2) p(s^2)$$

We are given the prior for D (with some very mangled notation and missing close-brackets) but we have to guess at the likelihood (something to do with the binomial distribution specified above?) and the prior for σ (which I wrote as s above for brevity). More deeply, it seems very unsatisfactory that the 'data' here, i.e. n , q , and α , aren't really data at all but outputs from other models (PyRate). It's not clear why this couldn't have been fitted in one step and just have one single overall model which explains the lot.

The entire section describing the mcmcDivE model has been rewritten for clarity, again double checking for any missing notation (lines 630-686).

Estimation of preservation rates are made computationally tractable by PyRate and so the preservation rates used in mcmcDivE are taken from the highly accurate model output, rather than calculated *de novo*. However, the mcmcDivE also uses the same fossil occurrence data input as PyRate and so the sampled-diversity portion of the model is informed by empirical data, rather than the output of another model.

The reviewer is correct that the mcmcDivE parameters could be estimated as part of a standard PyRate analysis and we note this in the text (lines 686-687). However, performance may be poor for large datasets such as the one we analyse, and integration of the mcmcDivE model with the PyRate model is not the goal of this paper. Given the already substantial complexity of the PyRate analysis, we believe that it is also clearer to present and implement the mcmcDivE model separately at this stage.

As a final minor point, it seems a tad strange for such a simple model (note I am not criticising the model for being simple - I like simple models) not to have been fitted in one of the many standard tools (Stan, JAGS, Nimble, many of which have both Python and R versions) to save them the trouble of coding this all up by hand. My guess is that a Hamiltonian Monte Carlo version of this model you would need only a few thousand iterations and nothing like the million plus samples that the authors use here.

As the reviewer states, this is a minor point. Implementation as a standalone Python program will ease the uptake of the mcmcDivE method, rather than provision as a script for other MCMC software. Further, as it uses the output of PyRate analyses and can provide outputs compatible with PyRate plotting functions, it is also convenient for the script to be able to run in the same overarching environment as PyRate, rather than requiring the user to move between multiple software platforms to perform the analysis.

Reviewer #4 (Christopher Dean):

The Carnian-Pluvial Event has attracted a growing amount of attention within recent years, and as such a quantitative study assessing the potential impacts of the CPE on marine and terrestrial ecosystems is a more than welcome addition to the literature. The authors perform an extensive list of tests with the aim of resolving the effects of the CPE on diversity, extinction and origination rates and ecological networks. They conclude that the Mid Carnian Biotic Crisis resulted in enhanced turnover and large changes in community ecology within both marine and terrestrial settings. Whilst this topic is well suited for the remit of this journal, there are a number of serious concerns I have regarding both the interpretations of the data that prevent me from recommending it for publication at this time.

We are pleased that Dr Dean endorsed the importance of the original manuscript and hope that the topic remains of interest to Nature Communications after the substantial revisions we have made. Those revisions address the two major criticisms raised, namely the interpretation of the results at the Carnian Pluvial Episode and the issue of geographic sampling bias, and the latter issue is now the key focus of the paper itself.

The authors carry out a great number of tests to quantify the effects of the Carnian Pluvial Event. However, many of these tests, in my opinion, do not show the the Carnian Pluvial Event as being a major mass extinction and I do not feel that the specific interpretations of the impacts of the CPE are backed up by the data presented. In almost all of the figures presented for this paper, equally large or larger changes are observed in either the preceding or following time bins. For example, an equally large drop in marine diversity is found within the Early Carnian (~243 mya) in probabilistic and traditional SQS methods; larger changes in all network analysis approaches are observed between 250-240 mya for both marine and terrestrial organisms; and extinction and origination rates show extinction drastically outpacing origination within the Late Norian and Rhaetian, to a degree that is not seen across the Carnian. In fact, both probabilistic and traditional diversity metrics report an increase in terrestrial diversity across the CPE. If I was shown the various data presented in this study without prior knowledge of the CPE, I would be more interested in examining the events before and after this event, as they show greater volatility and change. Whilst the authors acknowledge that there appears to be a lack of impact on terrestrial ecosystems and comment that 'the taxonomic and ecological effects of extinction are just as important as the magnitude', there is no methodological attempt to dig into why there might be a disparity here. Additionally, the following argument that terrestrial networks were significantly impacted across the CPE which is highlighted within the abstract ignores the fact that the drop in network transitivity is caused by a fall from an anomalous spike in the Early Carnian, before the CPE, and is returning to pre-spike levels. To me, this pattern is more interesting to following than any changes across the Early-Late Carnian boundary. As such, it appears to me that the authors are already convinced of the CPE as an event that caused major impacts within ecosystems, and are selectively interpreting data to fit this narrative. This also makes it very hard to comment on the discussion, which builds off the authors' interpretations. This paper would be far stronger if the data was presented neutrally and reasons for disconnect between this and prior studies were dug into more deeply.

Rather than focusing on the signal of the Carnian Pluvial Episode within the context of the Triassic, we now focus on diversity and diversification rates throughout the Late Permian to Early Jurassic, using the much better sampled marine fossil record only. In this way we can compare macroevolutionary changes throughout the entire study interval, highlighting the End Permian and End Triassic mass extinctions and the Carnian Pluvial Episode to each other. We have ensured that the discussion is presented as neutrally as possible, with due consideration of the true pattern of macroevolutionary history versus the uncertainty in the data and results. Further, our regional approach has led us to completely revise our interpretation of the Carnian Pluvial Episode.

Another serious issue is related to the lack of investigation around other potential causes of the patterns observed here. Biases arising from spatial variation in sampling have increasingly been seen as important for understanding diversity and ecology in deep time. Changes in the amount and location of sampled outcrop between stages has the potential to be viewed as extinction events or major changes in ecological structure rather than a bias arising from geological and anthropogenic means. Consequently, there have been a multitude of papers within the last few years that discuss the spatial biases in the fossil record and how to counteract them (e.g. Benson et al. 2016, Close et al. 2017, Dean et al. 2019, Close et al. 2020a,b). These issues are not addressed or mentioned within this manuscript, which seems like a major oversight for a study that uses diversity curves. Whilst I understand that the probabilistic methods here intrinsically incorporate rates of sampling,

these can still be affected by the underlying change in location of the sampling pool. Currently, it is impossible to evaluate whether patterns like the supposed gap between northern and southern communities in the Late Carnian are a result of genuine ecological signals, or a lack of sampling opportunity between the locations. The same is seen for extinction intensities (Figure 4); it is hard to know whether the Late Carnian genuinely shows an increase in extinction intensity, or just an increase in sampled area compared to the other stages (or vice versa, whether the other stages would show equally as intense extinction intensities if it was possible to sample more cells). This section could be drastically improved by implementing a geographic subsampling approach that randomly selected a certain number of geographic cells, and I would be a lot more comfortable discussing the results if these kind of measures had taken place.

The papers by Close et al. 2020 make the full extent of spatial bias in the fossil record clear – these came out when the original manuscript was already in review, although we should have paid closer attention to the preceding literature on the topic. Mitigation of geographic sampling bias in such a way that produces datasets compatible with rate and diversity estimation (probabilistic or otherwise) is now the *raison d'être* of the revised manuscript. We paid careful attention to the methods utilised in the papers mentioned above in order to incorporate their findings and recommendations into our spatial subsampling procedure. In short, the method allows formal spatial standardisation protocols to be applied to data from any user-defined spatial region through time, allowing regional signals to be disentangled from each other as well as from geographic sampling bias. As mentioned above, this regional approach has led us to completely revise our interpretation of the Carnian Pluvial Episode, given the substantial geographic variability in how it manifested biotically.

There are also temporal biases at play. Although I am not the most familiar with this period of geological history, I know that obtaining accurate chronostratigraphic ages for formations and fossil occurrences at this time is problematic, and that there is a reasonable amount of stratigraphic uncertainty. In fact, the authors themselves mention this within the paper. Whilst the authors compiled new ages for formations in the dataset, this is not really discussed in the context of the results, and it is unclear what knowledge this gave the authors of the stratigraphic uncertainties associated with this time period. Discussion of this uncertainty and the limitations of the data would strongly improve the manuscript and give greater confidence in interpretations.

Temporal biases are certainly present, so substantial data compilation and cleaning was conducted to improve the signal-to-noise ratio and precision in our data. Firstly, we utilise a composite dataset nearly three times the size of that used in our original paper to improve spatiotemporal and taxonomic coverage of our study interval. We again improved stratigraphic age uncertainty through revision using the primary literature and provide a greater discussion of the effect of our cleaning and revision on data quality and its analytical limitations (lines 441-477). Ultimately, however, we are less concerned with stratigraphic uncertainty compared to previous publications as our probabilistic methods incorporate that uncertainty into the modelling procedures. As such, all results and their interpretation and discussion are performed with that uncertainty already accounted for.

As a final point, whilst I cannot comment on the validity of the new probabilistic diversity method in terms of a technical standpoint, I am almost in agreement with the previous reviewer who commented that it would be better the method be presented in a separate study. Whilst I do not agree that is necessary to have a completely separate study to implement a new technique, I do find it hard to appropriately assess the severity of the drops in relative probabilistic diversity when there is no appropriate comparison with other major extinction events. I would be more convinced about the severity of the Mid-Carnian Biotic Crisis if I could compare diversity curves with those run over the P-T extinction, or the T-J extinction. I understand this would potentially require a substantial amount of additional work, but it would provide good evidence of the impact, or lack of impact of the CPE.

Our expanded study interval now fully encompasses the PT and TJ extinctions, permitting explicit comparison of all three events (or any point within each study region throughout the study interval more generally).

I don't want these general comments to be entirely negative – the authors have done a great job tying together a number of complex, intricate analyses carried out on an enormous dataset, and the manuscript itself is clear and well written. The extent of the supplementary data also shows the lengths that the authors have gone to to carry out clear and well documented science. Additionally, whilst my knowledge of Bayesian approaches is limited, the newly documented method for diversity estimation looks interesting, and is sure to be something a great number of palaeontologists and ecologists will be interested in. As such I feel this work fits well within the remit of Nature Communications. However, I feel that the work cannot be published without a more balanced discussion of the data presented and its limitations, and without additional measures taken to account for spatial biases in the fossil record. My overall recommendations for the authors would be to simplify the study, and focus on a few results (for instance the probabilistic diversity curves with the addition of spatial sampling) in greater depth.

We again thank Dr Dean for his positive outlook on the original submission. We believe that our revised manuscript fully meets his overall recommendation as stated above – it was a challenge, but ultimately a fun and informative one.

Additional relevant points (all Christopher Dean)

To my knowledge, the Forbes index compares species-site matrices to estimate the similarity or difference between ecological communities. In the figure caption you say Figure 5A shows “Change in Forbes2 ecological similarity”. In the methods section, you show say Forbes2 is used to calculate “Changes in ecological similarity between successive pairs of time bins throughout the Triassic”. If this is the case, what are the scores on the figure supposed to show? Are the scores within each bin the comparison between the and the previous bin? To me it appears as if a time series graph is not the best method to present these data. I would recommend a similar approach to Brocklehurst et al. (2017) instead.

However, more broadly I feel there is a potential issue with how Forbes2 is being implemented here. You state within the methods that you binned occurrences within all time bins that they potentially appeared within due to stratigraphic uncertainty. However, you are now using Forbes2 to compare ecological similarity between successive time bins. In which case, are you increasing overall ecological similarity between stages which show a greater degree of stratigraphic uncertainty due to the same occurrences being placed in both bins? This issue needs to be clarified and addressed before these results can be properly analysed.

While the figure comment is no longer relevant, we still utilise the modified Forbes index additionally with relative abundance correction (RAC; Brocklehurst et al 2018). The comparison is made between successive time bins (Fig 3), we have made sure this is clear in the new figure caption and methods section, and we present the data with the approach used in Brocklehurst et al (2018) where the RAC-Forbes method was described. Regarding the binning of occurrences, we have still used our binning approach. While this will increase the similarity between successive bins to a degree, we believe that it is the approach which best reflects the uncertainty in the data and despite the potential for increased similarity, there are still clear trends in turnover through time in each study region. We also note that the RAC-Forbes method accounts for additional uncertainty in the similarity measurement arising from incomplete sampling and uneven abundance distributions, making it more robust than Forbes alone.

Age constraints for formations are an incredibly important part of palaeontology that is often forgotten, so I am glad to see this being carried out. However, it would be good to have some additional information. Were these ages updated to have specific ages for the starts

and ends of formations? Or were they updated to fit within the time bins chosen within this study? I had a look in supplementary information and could not find a record of a list of formations, so I assume it is the latter. If so, it would be interesting to know a breakdown of how many formations can be actually split into the Early or Late Carnian, and how many cross over that interval.

How many occurrences were retained that had poor time constraints? How badly were the data affected? You have already made reference to the fact that tightly controlled time intervals are important. Providing these kind of confidence intervals is crucial for readers to be able to accurately interpret your data.

Ages were updated to substage-level time bins, then absolute numeric ages assigned from GTS2020 – documented in the methods. We also provide a brief discussion of the impact of this redating with regard to resultant age precision of the occurrences. We also imposed a cutoff of 10 Ma age uncertainty above which data was dropped. This had only a minor effect on the amount of data taken forward to analysis – again now documented in the methods. In general, virtually all of our data is at substage level, with 3-5 Ma uncertainty, and that remaining uncertainty is incorporated into all Bayesian modelling of diversity and diversification rates.

Why quorum of 0.8? This seems unusually high for a deep-time diversity study, and I am surprised you can get accurate curves from this. Did you attempt to see patterns at lower quora?

While this comment is no longer relevant in its original context, we made sure for the new manuscript to use more conservative quora for all relevant analyses. RAC-Forbes analyses use a very typical standardisation threshold of 0.5, while the SQS diversity estimates used to interrogate the effect of spatial extent on diversity were assessed at thresholds of 0.4, 0.5, 0.6 and 0.7 (all standard thresholds used in the literature) to incorporate any sensitivity to standardisation threshold into our analyses.

Please excuse this if I have misunderstood the method, but I have a slight issue with the high temporal resolution used here. From my understanding, the method calculates the relative diversity at each 1 ma time bin of a total 50 time bins based on the fossil occurrences present and a calculated sampling probability. However, on a check of the full dataset in R, there are only about 15 unique ages associated with the dataset. Isn't there therefore a large discrepancy between the resolution of the analysis carried out in comparison with the occurrence data available? I think my concerns would be lessened if you presented an overview of the temporal limits of formations and occurrences, so that I could see the comparison between the different resolutions.

As mentioned previously, we now provide greater discussion of the age uncertainty in our data in the methods. The vast majority of our data is at a substage level, around 3-5 Ma. For the relative diversity curves from mcmcDivE, the curve resolution is certainly higher at 1 Ma intervals, but this is the advantage provided by estimating the diversity curve from the model, parameterised by the empirical data, compared to bin-wise estimation directly from the empirical data itself. While at a higher resolution, the diversity curve still incorporates the age uncertainty in the original data with the confidence intervals derived from the posterior density of the diversity estimates.

As a general comment, it would be extremely helpful to have the stage designations, or chosen bins that are discussed in methods on graphs, and not just the grey boxes indicating stages. It is currently difficult to assess which bin is which bin without going back and forth between the figures and the methods. This extends to Figures 1, 2, 3 and 5.

While the figures are now different, we have made sure to include properly labelled chronostratigraphic timescales on the main text figures, along with grey boxes to permit easy comparison.

For Figure 3, the y axis on graphs B and F says Speciation Rate, whereas the figure caption says origination rate. This might cause some confusion in readers – I would suggest consistency or stating somewhere in the manuscript that these terms are used interchangeably for this work.

While the figures are now different, we have ensured that we have used origination rate consistently throughout the manuscript rather than speciation rate, given that our analyses were run at the genus level rather than the species level.

References

Brocklehurst, N., Day, M. & Frobisch, J. Accounting for differences in species frequency distributions when calculating beta diversity in the fossil record. *Methods Ecol. Evol.* 9, 1409-1420 (2018).

Close et al. The apparent exponential radiation of Phanerozoic land vertebrates is an artefact of spatial sampling biases. *Proc. R. Soc. B* 287, 20200372. (2020a)

Close, R., Benson, R., Saupe, E., Clapham, M. & Butler, R. The spatial structure of Phanerozoic marine animal diversity. *Science* 368, 420-424 (2020b).

Silvestro D., Schnitzler J., Liow L.H., Antonelli A. & Salamin N. Bayesian estimation of origination and extinction from incomplete fossil occurrence data. *Syst. Biol.* 63, 349-367 (2014).

Silvestro D., Salamin N., Antonelli A. & Meyer X. Improved estimation of macroevolutionary rates from fossil data using a Bayesian framework. *Paleobiology* 45, 546-570 (2019).

Warnock, R., Heath, T. & Stadler, T. Assessing the impact of incomplete species sampling on estimates of origination and extinction rates. *Paleobiology* 46, 137-157 (2020).

Reviewers' Comments:

Reviewer #3:

Remarks to the Author:

The authors' response to the main queries I had about the model and the notation in the last round has been to remove the vast majority of the mathematics and so leave it up to the reader to find the other papers behind much of the work here. I think this means that the abstract needs to be changed slightly, as it seems that the 'state of the art Bayesian methods' are actually in other papers, and are not part of the present paper's novelty. Perhaps the other parts of the paper presented here are worthy of publication in Nature Communications. However they are not in my area of expertise.

I did have a closer look at the methods and materials section. I still found it slightly confusing, as the first sentence suggests that temporal continuity is a desirable property of the output, but this achieved by binning (i.e. discretising) the time variable. It wasn't entirely clear to me why everything needed to be discretised as this will necessarily lose information, but again perhaps this is clear to those who work in this area. It does seem like this part is crying out for some mathematics to explain to the user.

The remaining model left in the paper (in the section Probabilistic Diversity estimation) is well presented and quite interesting. As I stated in my previous review it looks very much like an N-mixture model used commonly in Ecology. Some of the formatting seems to have gone wrong, for example missing parentheses in L649, and I don't think the square root should be in the second argument of the equation in L668.

Reviewer #5:

Remarks to the Author:

General comments

- This study presents a new approach for estimating diversity and diversification rates within distinct geographic regions through time and applies this to Late Permian to Early Jurassic marine fossil data.
- In my opinion, this study does some very interesting things and has the potential to be a great paper worthy of publication in Nature Communications. (Although I should declare that I am not sufficiently familiar with the workings of Bayesian methods like PyRate and LiteRate to be able to properly assess the methodological decisions pertaining to these tools.)
- However, I think that the way that the nitty-gritty details of the results are presented and conveyed could be improved quite substantially. In its current form, the main-text Results and Discussion sections lean heavily on citing long sequences of supplementary figures and tables, which all show similar sets of patterns for numerous different regions but which have minimal additional information for aiding interpretation.
- In my opinion, it is very difficult to relate statements made in the main text about the patterns in the results to these numerous separate supplementary figures, which lack geological/stratigraphic timescales or any other frame of reference to identify key biotic and environmental events (e.g., the Carnial Pluvial Event) that the results text makes reference to.
- I think that there should be some carefully-thought-out figures that summarise the key results in these sets of supplementary figures across all regions and allows them to be efficiently compared, so that the reader doesn't have to cross-correlate patterns across numerous figures. I worry that in the current form of the MS, most readers hoping to understand your interpretations of results will take a brief look at the numerous un-annotated figures in the supplement and give up.
- These figures — in fact, all of the figures, including those from the main text — also need to be annotated in ways that convey the key messages better and guide the reader's understanding, particularly in terms of identifying the biotic and environmental events that are referred to in the text (e.g., the CPE). At the moment there's no frame of reference for the reader to understand how the numerous wiggly lines relate to these events, and that's a pretty major shortcoming.

- On a related note, I think that the main text needs one of these summary figures showing origination and extinction rates across all regions (plus informative annotations to aid interpretation, as suggested above). This is important because these results are central to the findings, and are singled out in summaries of the results (e.g., in the abstract: "origination and extinction rates are regionally heterogeneous even during events that manifested globally"). At the moment they're spread across numerous supplementary figures and difficult to parse.
- Continuing in the theme of making things easier for the reader (in terms of the reader doing less work to digest your messages), it would be helpful to give supplementary figures more descriptive/informative captions (i.e., summarise the key take-aways and the authors' interpretations of the results) so they can be interpreted by the reader more easily.
- L359: section "Spatial standardisation workflow". It would be helpful to include a figure explaining all the steps taken with manipulations of MSTs and sliding windows, etc. It's a bit hard to follow everything fully without visual aids.

More detailed or specific comments

- L49: Ref 17 should be Ref 2?
- L86: at some point in the text it might be worth noting that Close et al. (2020, Science, Fig. S5) presented a somewhat similar sliding-window approach. The analysis summarised in their Fig. S5 traced diversity for specific geographic regions through sequential intervals of geological time (i.e., regions with sampling that allowed faunas to be traced from one time bin to the next). I think their method was based on quantifying the degree of spatial overlap of spatially-standardised samples in adjacent time bins.
- L92: "Figs. S1–S6". Should this actually be S1–S7? It's a bit confusing because there are duplicate supplementary figure numbers in the supplement files.
- L108: "Figs. S8–S14". I think the supplementary figure numbers are wrong in file 217480_3_supp_6139696_r448g8.pdf (S1–S7 used instead of S8–...).
- L109: "S1–S64". I'm not sure providing 64 supplementary tables is the best way to present this information — I'm concerned that readers will be overwhelmed by the amount of information and never look at these tables. Can the information be summarised in a more informative figure instead?
- L125: "Figs. 2, 3" I think the focus of the interpretation of results in the main text should be on the main-text figures, but Fig. 2 is only mentioned in passing. It would be useful to annotate Figs 2 and 3 with major events to aid the reader's interpretation and link features with things mentioned in the Results.
- L154: "As with alpha diversity..." Needs pointer to section heading? This sentence seems to imply that alpha was discussed earlier in the MS.
- L218: The same goes for other main-text figures, but Fig. 4 would particularly benefit from interpretative annotations to more effectively communicate the authors' preferred narrative to the reader.
- L233: "consonant" — do you mean consistent?
- L244: "Although this pattern may be influenced by the change in the spatial extent of the data..." Is there any way you could rule out changes in spatial extent as an explanation for massive spikes in origination in North Panthalassic region immediately after PTME?
- L271: "it may instead be the case that the stage-level resolution of many of the occurrences in each region is driving this signal" If PyRate picks up these kinds of artefacts, should stage-level occurrences be used at all to estimate patterns at finer temporal resolution? Could you get around these binning artefacts by running the analysis at stage-level?
- L285: "the crash in spatial extent" Can you have a crash in spatial extent? How about "sharp contraction in spatial extent" or similar?
- L293: "In the Tangaroan region, however, beta diversity declines across the event, showing that faunal composition of the region changed little across the extinction boundary." I'm not sure I follow your logic here — how does a decline in beta diversity show that faunal composition in the region didn't change much? They seem to me to be two separate things.
- L311: "supporting those aspects of our results and further suggesting that . that the ecological" Stray full stop in the middle of the sentence.

- L389: "MSTs with different aspect ratios may show similar total lengths but could sample over very different spatial extents, inducing a bias by uneven sampling across spatially organised diversity gradients¹⁵". I think this should be ref 16, not 15?
- L425: There's a lot of scope for confusion if you apply the term "alpha diversity" for diversities of larger regions. In the fossil record it's more commonly used to refer to per-collection or local-scale diversity.
- L520: "40%, 50%, 60% and 70%." To avoid biasing diversity with signals of evenness, you should try to maximise coverage used. Even 0.7 is quite low, and in my opinion values lower than that shouldn't be reported. You should usually choose the highest value achievable for all assemblages you want to include. Can you not get away with using significantly higher levels, like 0.9?
- L697: I'm not certain from reading the text exactly what beta diversity was calculated for. Was each region traced through time treated as a sampling unit ("alpha") and beta calculated across all regions? Or is beta calculated within each region? The main text figure showing beta curves seems to imply the latter. But then how was alpha calculated?
- L630–638: "The method estimates two classes of parameters: the number of unobserved species for each time bin and a parameter quantifying the volatility of the diversity trajectory." Given that this is a brand-new method for estimating diversity, I think that it would be very useful to see a comparison between the diversity trajectories estimated by this new method, mcmcDivE, and existing methods for estimating sampling-standardised diversity, like SQS and extrapolators like Chao 1. Figures overlaying (normalised) time series and bivariate plots would be useful. The interesting thing to know is how the results of the methods differ (and why?). It's a shame that this interesting new method has been shoehorned into the supplement of this paper. In my opinion it would have been preferable to give it its own paper in order fully explore how it works and performs relative to existing methods. But at a bare minimum it would be good to show how it performs compared to SQS and non-parametric richness extrapolators like Chao 1.

Response to reviewers

Reviewer comments are given in regular text, our responses are in red.

Reviewer #3:

The authors' response to the main queries I had about the model and the notation in the last round has been to remove the vast majority of the mathematics and so leave it up to the reader to find the other papers behind much of the work here.

We made a conscious decision to remove a number of the previous equations so that the remaining mathematics relates entirely to the mcmcDivE model, the new Bayesian method presented in the manuscript. Firstly, this allowed us to present the mcmcDivE model with greater clarity (in response to the reviewer's remark on the previous round of review, and successfully, judging by their later comment here). Secondly, the mathematics which we removed relates to the preservation model at the core of PyRate. This is already published and is best explained in the original papers presenting PyRate, so we believe that it is more appropriate to direct a reader to these sources with our reference list should they wish to know more about the method, rather than rehashing parts of the preservation model in our manuscript in a manner which only made presentation of the mcmcDivE model needlessly complicated.

I think this means that the abstract needs to be changed slightly, as it seems that the 'state of the art Bayesian methods' are actually in other papers and are not part of the present paper's novelty. Perhaps the other parts of the paper presented here are worthy of publication in Nature Communications. However, they are not in my area of expertise.

As a point of semantics, 'state of the art' does not necessarily mean complete novelty. PyRate methods are presented in previous papers, although our workflow coupling PyRate and LiteRate to permit analyses of large datasets is new and we still view both methods as 'state of the art' given that nothing more advanced has come along since, and that they are still being refined to incorporate ongoing advancements in the field – i.e., they continue to remain at the forefront of their broader methodological toolbox. The mcmcDivE model for probabilistic diversity estimation is categorically a novel Bayesian method. We regard the use of PyRate and (novel) mcmcDivE to be key strengths of how we have approached the empirical problems addressed by our manuscript. To avoid semantic issues, however, we have revised the abstract to better reflect where the methodological aspects of our manuscript sit with regard to current methods (L12–14).

I did have a closer look at the methods and materials section. I still found it slightly confusing, as the first sentence suggests that temporal continuity is a desirable property of the output, but this achieved by binning (i.e. discretising) the time variable. It wasn't entirely clear to me why everything needed to be discretised as this will necessarily lose information, but again perhaps this is clear to those who work in this area. It does seem like this part is crying out for some mathematics to explain to the user.

We have revised our phrasing to 'consistent' (L375, L389) to avoid the issue of continuous vs discretised data. The idea is that the geographic distribution of the subsampled data remains consistent through time, even if those measurements are from discrete time bins. For workers in the subject area, discretisation into time bins when working with fossil occurrence data is a known necessity. This discretisation is only required by the spatial standardisation method, however, as the diversification rate and diversity estimates are model-derived, and results can be generated at any user-defined time-step. The mathematics is given in the referenced papers for PyRate, and for mcmcDivE in the Materials and Methods.

The other reviewer also picked up on the issue of clarity with our spatial standardisation method (which this comment appears to relate to in the manuscript). As stated below, we have substantially clarified our presentation of the standardisation method with an additional figure. Otherwise, there are not really any mathematics nor models to present here. The

spatial standardisation procedure is entirely algorithmic and the only specific mathematical components they utilise are great circle distance and correlation tests, the formulas for which we have never seen included in the Materials and Methods sections of other palaeontological papers employing these tools.

The remaining model left in the paper (in the section Probabilistic Diversity estimation) is well presented and quite interesting. As I stated in my previous review it looks very much like an N-mixture model used commonly in Ecology.

We are glad that this section is now clear to the reader. We have additionally moved our simulation-based validation of mcmcDivE into the main text within the methods section as these analyses are central to demonstrating its strength (L730-788, Fig. 9). We agree that our model shares some similarities with N-mixture models used in ecology in that it estimates a latent diversity or abundance count with autocorrelated priors. However, our approach differs in the underlying sampling process (e.g. accounting for rate heterogeneity through time and across lineages) in how preservation is inferred (through a Poisson process with a birth-death prior), and in the use of a Brownian process prior for diversity through time.

Some of the formatting seems to have gone wrong, for example missing parentheses in L649, and I don't think the square root should be in the second argument of the equation in L668.

Parentheses were indeed missing, correction made (L689). We are positive that the equation (L710) is corrects

Reviewer #5 (Remarks to the Author):

- This study presents a new approach for estimating diversity and diversification rates within distinct geographic regions through time and applies this to Late Permian to Early Jurassic marine fossil data. In my opinion, this study does some very interesting things and has the potential to be a great paper worthy of publication in Nature Communications. (Although I should declare that I am not sufficiently familiar with the workings of Bayesian methods like PyRate and LiteRate to be able to properly assess the methodological decisions pertaining to these tools).

We are glad that the reviewer endorses the importance of our work. Regarding the Bayesian methods, we developed our workflow and analytical strategy in discussion with the author of PyRate and made every effort to follow what established practices already exist in the literature for PyRate, as well as for Bayesian methods in general (e.g. assessing convergence using ESS).

- However, I think that the way that the nitty-gritty details of the results are presented and conveyed could be improved quite substantially. In its current form, the main-text Results and Discussion sections lean heavily on citing long sequences of supplementary figures and tables, which all show similar sets of patterns for numerous different regions, but which have minimal additional information for aiding interpretation.

We address specific sets of results in response to the reviewer's comments below. Broadly, however, our solutions have been the creation of new figures to summarise the key information in the supplementary figures, augmentation of existing figures to make their relationship to the discussion points clearer, minimising reference to the long series of supplementary tables and figures in the text, and improvement of figure captions so that their take-home messages are clear. We retain the original supplementary files so that the complete information is available to a reader who wishes to delve deeply into the differences between data treatments and regions, but we ensure that the key results of our manuscript are not reliant on the reader having to compare trends across multiple sets of supplementary figures and are instead presented clearly and efficiently by the main text and figures.

- In my opinion, it is very difficult to relate statements made in the main text about the patterns in the results to these numerous separate supplementary figures, which lack geological/stratigraphic timescales or any other frame of reference to identify key biotic and environmental events (e.g., the Carnian Pluvial Event) that the results text makes reference to.

We have removed, as much as possible, broad statements which were supported by references to large series of supplementary figures as they deducted from the clarity of the results and discussion. The supplementary figures mostly record the impact of our standardisation methods on the data, from which we have selected the most effectively standardised results to present as figures in the main text. These main text figures have been updated so that the biotic/environmental events discussed can be clearly identified (see next comment). We still make some references to the supplementary figures, but we keep this as specific as possible, e.g. Fig. S6, rather than Figs. S1-S7, again so that comparison across multiple supplementary figures is not necessary to digest the findings of the manuscript.

- These figures — in fact, all of the figures, including those from the main text — also need to be annotated in ways that convey the key messages better and guide the reader's understanding, particularly in terms of identifying the biotic and environmental events that are referred to in the text (e.g., the CPE). At the moment there's no frame of reference for the reader to understand how the numerous wiggly lines relate to these events, and that's a pretty major shortcoming.

We have added labelled, coloured bars to all time series figures, highlighting the chronostratigraphic position of the key events we discuss in the text: the end-Permian mass extinction, the Carnian Pluvial Episode and the end-Triassic mass extinction.

- On a related note, I think that the main text needs one of these summary figures showing origination and extinction rates across all regions (plus informative annotations to aid interpretation, as suggested above). This is important because these results are central to the findings, and are singled out in summaries of the results (e.g., in the abstract: "origination and extinction rates are regionally heterogenous even during events that manifested globally"). At the moment they're spread across numerous supplementary figures and difficult to parse.

We have inserted a chart (new fig. 5) in the style of new figs 3 and 4 in the main text, showing the MST-standardised extinction and origination rates in each region and have made frequent references to this figure in the main text.

- Continuing in the theme of making things easier for the reader (in terms of the reader doing less work to digest your messages), it would be helpful to give supplementary figures more descriptive/informative captions (i.e., summarise the key take-aways and the authors' interpretations of the results) so they can be interpreted by the reader more easily.

Full figure captions have been given to each set of supplementary figures so that their take-home message is quickly conveyed without the need for extensive cross-correlation between regions and data treatments (L1139-1187).

- L359: section "Spatial standardisation workflow". It would be helpful to include a figure explaining all the steps taken with manipulations of MSTs and sliding windows, etc. It's a bit hard to follow everything fully without visual aids.

We have created a further main text figure to illustrate the major steps of our spatial standardisation workflow (new Fig. 8), in summary: 1. Demarcation of the geographic region. 2. Translation through time to capture data. 3. Fixation of longitude-latitude extent of the data in each time bin, followed by spatial binning. 4. Standardisation of minimum spanning tree length. This figure is referenced throughout the description of the workflow (L378, L395, L426, L430).

More detailed or specific comments

- L49: Ref 17 should be Ref 2?

The references here all appear correct. Close et al 2020a (2), Vilhenna and Smith 2013 (15), Close et al 2017 (16) and Close et al 2020b (17) all relate to palaeogeographic bias in the fossil record.

- L86: at some point in the text it might be worth noting that Close et al. (2020, Science, Fig. S5) presented a somewhat similar sliding-window approach. The analysis summarised in their Fig. S5 traced diversity for specific geographic regions through sequential intervals of geological time (i.e., regions with sampling that allowed faunas to be traced from one time bin to the next). I think their method was based on quantifying the degree of spatial overlap of spatially standardised samples in adjacent time bins.

Our method was actually developed in response to the method of Close et al (2020) and draws some inspiration from it, specifically the use of a minimum spanning tree (MST) with tree nodes corresponding to cells in a hexagonal grid. The Close method finds all spatially nested subsamples of hexagonal cells within the total minimum spanning tree in a given time bin (all occupied hexagonal cells connected). Diversity is then estimated from the pool of taxa within a given subsample, allowing subsamples with a consistent target MST length to be analysed for a spatially standardised result. The average diversity from a series of spatially standardised subsamples can then be taken, but there is no formal method to ensure that those subsamples overlap through time, fall fully within the bounds of a designated region, nor is there any precise geospatial relationship between a subsample in one time bin and the subsample in another.

The Close method effectively provides a forest of MSTs (i.e. a spatial point cloud) where each MST is spatially standardised and from which an average diversity estimate can be taken, but no way to ensure that the spatial distribution of the entire forest remain consistent through time. While it is definitely useful, conceptually elegant, and more flexible than our approach for regionalised diversity estimation, this was the key limitation we identified and the one which hinders diversification rate estimation. It does not produce a single dataset, rather a series of discontinuous, bin-specific datasets which record fossil taxon occurrences, which cannot then simply be concatenated as the spatial extents of each forest of MST are not standardised (despite each individual MST being so), even when MSTs are assigned to a specific geographic region (i.e., continents in Fig. S5) or to a particular latitudinal band (as in the main text).

This is what our method instead achieves. Rather than standardising a series of MSTs in each bin, then assigning them to a region without further standardisation of the entire set (the Close method), our method first defines the region using a sliding window so that the area and extent of that region remains consistent through time (standardisation type 1). A single MST is then built within each iteration of sliding window (each time bin) to a target length (standardisation type 2), whilst forcing it cover the longitude-latitude extent of the sliding window (standardisation type 3). The data from each MST can then be concatenated into a single dataset as they all relate to the same spatially standardised geographic region (the sliding window), thus subsampling a consistent geographic area, consistent tree length and consistent longitude-latitude extent. This dataset will record fossil taxon durations through the temporal extent of the study region, unaffected by spatial sampling bias, as shown by our correlations with sampling-corrected diversity before and after spatial standardisation.

We provide a shortened version of this comparison in our materials and methods section to highlight the relationship between our method and that of Close et al 2020 (L378–390), but stress that they are ultimately very different methods, sharing only the elements of minimum spanning trees predicated on (hexagonal) spatially binned data.

- L92: "Figs. S1–S6". Should this actually be S1–S7? It's a bit confusing because there are duplicate supplementary figure numbers in the supplement files.

Yes, the range should be S1–S7 and we have updated this. We have also updated the supplementary figure numbers in the PDFs for S8–S14 to be correct.

- L108: "Figs. S8–S14". I think the supplementary figure numbers are wrong in file 217480_3_supp_6139696_r448g8.pdf (S1–S7 used instead of S8–...).
The supplementary figure numbers in the PDF have been updated to be S8–S14.

- L109: "S1–S64". I'm not sure providing 64 supplementary tables is the best way to present this information — I'm concerned that readers will be overwhelmed by the amount of information and never look at these tables. Can the information be summarised in a more informative figure instead?

We agree that these results needed refining. Originally, we treated the correlations something like a nuisance parameter – our method only had to ensure non-significant correlations between diversity and spatial extent after standardisation, and so the correlation results for each dataset were largely a means to the end goal of estimating diversity dynamics. They are rather dense though, as the reviewer points out, so we have created a figure (new fig. 2) for the main text which summarises the same information, highlighting correlation strength and the significance of those correlations with colours so that the differences before and after standardisation are presented more succinctly. We retain the supplementary tables so that precise values for each test are available, following best reporting practise, but direct the reader to the main figure as this clearly demonstrates the desired outcome of the standardisation procedure, along with Table 1.

- L125: "Figs. 2, 3" I think the focus of the interpretation of results in the main text should be on the main-text figures, but Fig. 2 is only mentioned in passing. It would be useful to annotate Figs 2 and 3 with major events to aid the reader's interpretation and link features with things mentioned in the Results.

As the main events we discuss are the end-Permian mass extinction, Carnian Pluvial Episode, and the end-Triassic mass extinction, we have added coloured bars demarcating the duration of these events on figures 2 and 3 so that the dissonance between diversity dynamics in each region is clearly highlighted. We have also added further figure references in the main text to the specific regional data (e.g. Fig 2C for West Circumtethys diversification rate) so that the figures are more strongly linked to the text and to better aid the reader in identifying the features of the curves referenced in the discussed.

- L154: "As with alpha diversity..." Needs pointer to section heading? This sentence seems to imply that alpha was discussed earlier in the MS.

We have qualified that our initial results for diversity are specifically for alpha diversity (now gamma in our revised manuscript – see comments below) (L115, L161 – number of taxa), so that this is clearly distinguished from our references to beta diversity elsewhere (degree of taxonomic differentiation).

- L218: The same goes for other main-text figures, but Fig. 4 would particularly benefit from interpretative annotations to more effectively communicate the authors' preferred narrative to the reader.

All figure captions (main text and supplementary) have been expanded so that their take-home messages are clearer (L1051–1187) and the addition of bars to highlight the timing of key events allows the differences between rate and diversity curves during these events to be clearly visualised.

- L233: "consonant" — do you mean consistent?
Change to 'consistent' made, L242

- L244: "Although this pattern may be influenced by the change in the spatial extent of the data..." Is there any way you could rule out changes in spatial extent as an explanation for massive spikes in origination in North Panthalassic region immediately after PTME?

Standardisation of the Triassic-Jurassic bins to the extent seen in the Permian would result in extreme data loss through the vast majority of the study interval, degrading the diversification signal. We detail in the results why we chose a standardisation target that gives us a trade-off between good quality results for the rest of the dataset and the potential issue of changing spatial extent across the PTME. While this is unfortunate, it cannot be avoided using our method, and we already use this as a case study of both a limitation of the method and a reality of the geographic structure of the fossil record through time. The standardisation method of Close et al would allow spatially standardised diversity samples across the PTME for North Panthalassa to be produced, but the differences in diversity would still not show the underlying balance of extinction and origination rates, the methodological issue we outline in our introduction. As discussed in the text, however, we posit that section-level stratigraphic analyses can overcome this limitation, and the high diversity in North Panthalassic fossil assemblages in the aftermath of the PTME supports the idea that the spikes in origination reflect biological process rather than sampling bias (L256–L259).

- L271: "it may instead be the case that the stage-level resolution of many of the occurrences in each region is driving this signal" If PyRate picks up these kinds of artefacts, should stage-level occurrences be used at all to estimate patterns at finer temporal resolution? Could you get around these binning artefacts by running the analysis at stage-level?

We realised that we did not explain this as well as we could have and have reworked the text accordingly (L281–286). Fig. S44 shows that most of our occurrences are constrained to substage level, so stage-level occurrences are not a major concern. However, many substage boundaries will still coincide with stage boundaries, and so substage-level FADs and LADS may still have the potential to induce a signal of biotic change across stage level boundaries. We also note that this section refers to the results from mcmcDivE, rather than PyRate – the PyRate results appear unaffected by stage boundaries. As the mcmcDivE model estimates are at a regular time step, a stage-level analysis is not really possible with the current implementation of the method, so a temporally coarser analysis would simply shift the hypothesised bias to coincide with coarser set of interval boundaries without any analytical gain. We stress that this is only a hypothesis and one that requires further investigation. This is beyond the bounds of this manuscript, however, and does not impact the main message of the paper that diversity dynamics are regionally heterogenous, with that heterogeneity showing up regardless of whether boundaries are stage or substage-level.

- L285: "the crash in spatial extent" Can you have a crash in spatial extent? How about "sharp contraction in spatial extent" or similar?

Change to 'sharp contraction in spatial extent' made, L298

- L293: "In the Tangaroan region, however, beta diversity declines across the event, showing that faunal composition of the region changed little across the extinction boundary." I'm not sure I follow your logic here — how does a decline in beta diversity show that faunal composition in the region didn't change much? They seem to me to be two separate things. We measured beta diversity for a given bin as the degree of taxonomic dissimilarity between the taxon pools in the focal and preceding bins (see later comment for our chosen definition of beta diversity). The beta diversity measurement in a bin therefore gives a measure of the intensity of taxonomic turnover from the previous bin (0 = no turnover, all taxa shared; 1 = total turnover, no shared taxa). We have rephrased the lines about Tangaroan beta diversity at the end of the Triassic (L306–L309) to better reflect the fact that the low degree of change is relative. There was still turnover in the Tangaroan region across the TJME as beta

diversity > 0. However, beta diversity is higher for preceding time bins, so the magnitude of turnover through the late Triassic was greater than that taking place across the extinction boundary.

- L311: "supporting those aspects of our results and further suggesting that . that the ecological" Stray full stop in the middle of the sentence.

Errant full stop removed, L326

- L389: "MSTs with different aspect ratios may show similar total lengths but could sample over very different spatial extents, inducing a bias by uneven sampling across spatially organised diversity gradients¹⁵". I think this should be ref 16, not 15?

Indeed, a typo, reference changed to 16, L418

- L425: There's a lot of scope for confusion if you apply the term "alpha diversity" for diversities of larger regions. In the fossil record it's more commonly used to refer to per-collection or local-scale diversity.

We agree that alpha diversity is better suited to small-scale diversity collections and have changed from alpha to gamma to better reflect that fact that our diversity estimates are region-scale. We elected not to use 'region-scale' itself, however, given that we also discuss beta diversity and so it seems more logical to stay with the (alpha), beta and gamma specification of the type of diversity under discussion. In some instances, we talk about diversity in general terms and so do not specify alpha/beta/gamma, but we always qualify the type of diversity when referring specifically to our results.

- L520: "40%, 50%, 60% and 70%." To avoid biasing diversity with signals of evenness, you should try to maximise coverage used. Even 0.7 is quite low, and in my opinion values lower than that shouldn't be reported. You should usually choose the highest value achievable for all assemblages you want to include. Can you not get away with using significantly higher levels, like 0.9?

While it is true that lower levels of coverage will induce some bias from evenness, we would disagree that 0.7 is relatively low coverage. 0.7 is a common standard across fossil diversity analyses utilising SQS and our total range of 0.4–0.7 is the range recommended by the original paper on SQS (Alroy 2010). Even the most recent study which compared the efficacy of a wide range of diversity estimators (Close et al 2018; ref 14 in the manuscript), including SQS, found that estimated diversity patterns remain geometrically similar (as is also the case for our SQS diversity results across our coverage thresholds), and the authors additionally noted that even coverage of 0.9 still remains biased when species sampling is highly uneven. With regard to our results, estimation of diversity by SQS is only to provide a first pass means of determining whether spatial standardisation has successfully removed geographic bias on diversity from the dataset, not a means of examining diversity trends themselves. Further, smaller regions do show lower coverage values in some time bins and so we chose a range of thresholds that could be applied across all regions and time bins so that our standardisation protocol was fully consistent.

Alroy (2010). Geographical, environmental and intrinsic biotic controls on Phanerozoic marine diversification. *Palaeontology*, 53, 1211-1235

- L697: I'm not certain from reading the text exactly what beta diversity was calculated for. Was each region traced through time treated as a sampling unit ("alpha") and beta calculated across all regions? Or is beta calculated within each region? The main text figure showing beta curves seems to imply the latter. But then how was alpha calculated?

The reviewer is correct – alpha (now gamma) diversity is calculated through time by treating each region as a sampling unit, and beta diversity is calculated within each region. In our methods section and supplement, we fully explain how sampling-corrected gamma diversity

is calculated using the mcmcDivE model (L667-728), and we additionally clarify that beta diversity is region-specific (L814).

We use the broader definition of beta diversity (given in terms of turnover, rather than the original, narrow definition in terms of gamma and alpha diversity). We calculate beta diversity as the degree of dissimilarity between taxon pools for successive time bins within a given region (L796). The evolving definition of beta diversity is discussed in Anderson et al (2011), which we include as an additional reference in our methods section (L796, ref 90) to support our use of the turnover-based definition. While the multiple definitions still unfortunately result in confusion in the literature, they are all valid and have all found use in studies of the fossil record. We also clarify our definition in the results section (L115, 161).

- L630–638: "The method estimates two classes of parameters: the number of unobserved species for each time bin and a parameter quantifying the volatility of the diversity trajectory." Given that this is a brand-new method for estimating diversity, I think that it would be very useful to see a comparison between the diversity trajectories estimated by this new method, mcmcDivE, and existing methods for estimating sampling-standardised diversity, like SQS and extrapolators like Chao 1. Figures overlaying (normalised) time series and bivariate plots would be useful. The interesting thing to know is how the results of the methods differ (and why?). It's a shame that this interesting new method has been shoehorned into the supplement of this paper. In my opinion it would have been preferable to give it its own paper in order fully explore how it works and performs relative to existing methods. But at a bare minimum it would be good to show how it performs compared to SQS and non-parametric richness extrapolators like Chao 1.

We are glad that the reviewer finds the method of interest and agree that comparison with established methods is key to assessing its robustness. Previous comparisons of diversity estimators (Close et al 2018, ref 14 in manuscript) show that SQS is consistently the best of the current methods for estimating sampling corrected diversity, including the version of Chao and Jost (2012) which can perform both interpolation and extrapolation to meet the desired level of coverage. SQS strongly outperforms the Chao1 estimator (Close et al 2018), so we elected not to conduct comparison with this extrapolator. Instead, we add comparisons to SQS as it is the current standard for palaeontological diversity estimation, and so the benchmark to improve upon in essence, and with the extrapolator squares (Alroy 2018) which was published after the comparative work of Close et al (2018) but has fared favourably in simulation experiments since (Alroy 2020). We achieved this by applying SQS (same function we use in our spatial standardisation workflow, as in Chao and Jost 2012) and squares extrapolation to the sets of simulation datasets that we analysed previously. We find that mcmcDivE outperforms SQS and squares which, along with the congruence between mcmcDivE diversity curves and PyRate diversification rate curves, strongly bolsters the credibility of our new method. As the reviewer also points out, this is a brand-new method and it is important to show that it is trustworthy to be able to confidently present the results of our empirical application, so we have additionally moved the details of the validation with simulations into the main text within the Methods section (L730-791, Fig. 9). Alroy (2020). On four measures of taxonomic richness. *Paleobiology*, 46, 158 - 175

Reviewers' Comments:

Reviewer #5:

Remarks to the Author:

I am very happy with most of the changes made by the authors in response to my previous review.

In my opinion, there is only one thing that should be changed before the MS can be accepted for publication. This is the author's use of the terms alpha/beta/gamma.

The authors use "beta" to describe bin-to-bin faunal dissimilarity as a measure of turnover, which is not how most people would interpret the term. They wrote a section that seeks to justify this by arguing that the term beta has a broad meaning. I agree that this kind of bin-to-bin comparison of assemblage composition *is* basically estimating temporal beta diversity, but I also think that using the term "beta" in this context is unnecessarily confusing for the average reader. A better term would be "turnover" or "within-region turnover", while making it clear that you're measuring bin-to-bin turnover using Forbes similarity index rather than more traditional turnover methods.

On a related note, I personally think that "regional-scale diversity" is more descriptive and less prone to misinterpretation than "gamma diversity" — which someone could easily mistake for the diversity of the whole system (all regions summed) if they weren't reading carefully. That way you can entirely avoid all of the confusion and ambiguity arising from alpha/beta/gamma terminology.

Lastly, I noticed that the Figure 4 legend still says alpha diversity instead of gamma (if you follow my suggestion, this should be changed to "regional-scale diversity").

Response to reviewers

Reviewer comments are given in regular text, our responses are in red.

I am very happy with most of the changes made by the authors in response to my previous review. In my opinion, there is only one thing that should be changed before the MS can be accepted for publication. This is the author's use of the terms alpha/beta/gamma. The authors use "beta" to describe bin-to-bin faunal dissimilarity as a measure of turnover, which is not how most people would interpret the term. They wrote a section that seeks to justify this by arguing that the term beta has a broad meaning. I agree that this kind of bin-to-bin comparison of assemblage composition *is* basically estimating temporal beta diversity, but I also think that using the term "beta" in this context is unnecessarily confusing for the average reader. A better term would be "turnover" or "within-region turnover", while making it clear that you're measuring bin-to-bin turnover using Forbes similarity index rather than more traditional turnover methods.

We agree with the reviewer that the wide range of contexts to which 'beta diversity' has been applied creates strong scope for confusion. We have updated our phrasing to 'turnover' as suggested.

On a related note, I personally think that "regional-scale diversity" is more descriptive and less prone to misinterpretation than "gamma diversity" — which someone could easily mistake for the diversity of the whole system (all regions summed) if they weren't reading carefully. That way you can entirely avoid all of the confusion and ambiguity arising from alpha/beta/gamma terminology.

As we have removed any reference to beta diversity as per the previous remark, we no longer needed to distinguish between 'beta' diversity (= turnover) and 'gamma' diversity (= number of taxa). As such, we have also removed all references to gamma diversity and instead refer to just 'diversity' and 'turnover' throughout the manuscript. These measurements are all taken at a regional level, and we ensure that this is specified at appropriate points throughout.

Lastly, I noticed that the Figure 4 legend still says alpha diversity instead of gamma (if you follow my suggestion, this should be changed to "regional-scale diversity").

All figure legends have been checked and any references to alpha or gamma removed, as per the previous remark